# Variation in bridgmanite grain size accounts for the mid-mantle viscosity jump

Hongzhan Fei[1,2 ✉], Maxim D. Ballmer[3], Ulrich Faul[4], Nicolas Walte[5], Weiwei Cao[6] & Tomoo Katsura[1,7]

A viscosity jump of one to two orders of magnitude in the lower mantle of Earth at 800–1,200-km depth is inferred from geoid inversions and slab-subducting speeds. This jump is known as the mid-mantle viscosity jump[1,2]. The mid-mantle viscosity jump is a key component of lower-mantle dynamics and evolution because it decelerates slab subduction[3], accelerates plume ascent[4] and inhibits chemical mixing[5]. However, because phase transitions of the main lower-mantle minerals do not occur at this depth, the origin of the viscosity jump remains unknown. Here we show that bridgmanite-enriched rocks in the deep lower mantle have a grain size that is more than one order of magnitude larger and a viscosity that is at least one order of magnitude higher than those of the overlying pyrolitic rocks. This contrast is sufficient to explain the mid-mantle viscosity jump[1,2]. The rapid growth in bridgmanite-enriched rocks at the early stage of the history of Earth and the resulting high viscosity account for their preservation against mantle convection[5–7]. The high Mg:Si ratio of the upper mantle relative to chondrites[8], the anomalous $^{142}$Nd:$^{144}$Nd, $^{182}$W:$^{184}$W and $^{3}$He:$^{4}$He isotopic ratios in hot-spot magmas[9,10], the plume deflection[4] and slab stagnation in the mid-mantle[3] as well as the sparse observations of seismic anisotropy[11,12] can be explained by the long-term preservation of bridgmanite-enriched rocks in the deep lower mantle as promoted by their fast grain growth.

The lower mantle of Earth consists of bridgmanite as the most abundant mineral phase, followed by ferropericlase and davemaoite as the second and third phases, respectively. Silicate melting and solidification experiments[13,14] demonstrate that bridgmanite is the first phase to crystallize from a magma ocean in the early stages of the history of Earth. Owing to fractional crystallization[15], bridgmanite-enriched rocks with low ferropericlase proportion ($X_{fpc}$ <5–10%) were formed at more than about 1,000-km depth, evolving into pyrolitic (or peridotitic) rocks with relatively high $X_{fpc}$ (≈20%) at shallower depths, whereas the davemaoite content is lower than that of ferropericlase or even absent in the deep lower mantle[16]. The bridgmanite-enriched rocks could be preserved until the present day without mixing by mantle convection[5–7,17] as demonstrated by the current mantle seismic and density profiles, both of which agree well with pyrolitic compositions in the shallow lower mantle and bridgmanite-enriched rocks in the deeper regions[18–21]. A bridgmanite-enriched deep lower mantle is also supported by the density crossover between bridgmanite and ferropericlase—that is, bridgmanite-enriched rocks are denser than pyrolitic rocks in the mid-mantle[20].

It was previously considered that bridgmanite is rheologically stronger than ferropericlase[22–24]. Thus, bridgmanite-enriched rocks may have a higher viscosity than those of pyrolitic rocks, which may lead to an increase in viscosity with depth. The increase in strength of ferropericlase with pressure[23,25] and the iron spin transition[26] may also

cause an increase in viscosity. However, using these scenarios to explain an increase in viscosity of one to two orders of magnitude requires an interconnected framework of ferropericlase (ferropericlase-controlled lower mantle rheology)[5,22], which is unlikely because the electrical conductivity of the lower mantle is comparable to that of bridgmanite[27,28], but three orders of magnitude smaller than that of ferropericlase[27]. In particular, recent atomic modelling[29] shows periclase has a slower creep rate than that of bridgmanite under mantle conditions, whereas deformation experiments[30] suggest that bridgmanite has an identical creep rate to that of post-spinel (70% bridgmanite + 30% ferropericlase); both of these findings indicate a bridgmanite-controlled lower-mantle rheology. Moreover, the oxygen vacancies in bridgmanite formed by the substitutions of $Si^{4+}$ with $Al^{3+}$ and $Fe^{3+}$ have been proposed to cause an increase in bridgmanite strength with depth[31–33]. However, $Al^{3+}$ and $Fe^{3+}$ are more likely to form $FeAlO_3$ in bridgmanite[34]. Furthermore, the contribution of davemaoite to lower-mantle rheology should be limited as well because of its low volume fraction (and thus no interconnection)[16], although davemaoite is rheologically weaker than bridgmanite[35].

Because the viscosity ($\eta$) of polycrystalline aggregates has a strong grain-size ($d$) dependence ($\eta \propto d^2 - d^3$) in the diffusion creep regime, which may play an essential part in lower-mantle rheology[11], constraints on grain size and grain-growth rate of bridgmanite are crucial for understanding the viscosity of the lower mantle[36]. However, the grain size and grain-growth rate have so far only been experimentally investigated

[1]Bayerisches Geoinstitut, Universität Bayreuth, Bayreuth, Germany. [2]Key Laboratory of Geoscience Big Data and Deep Resource of Zhejiang Province, School of Earth Sciences, Zhejiang University, Hangzhou, China. [3]Department of Earth Sciences, University College London, London, UK. [4]Earth Atmospheric and Planetary Sciences, Massachusetts Institute of Technology, Cambridge, MA, USA. [5]Heinz Maier-Leibnitz Zentrum (MLZ), Technische Universität München, Garching, Germany. [6]Conditions Extrêmes et Matériaux: Haute Température et Irradiation (CEMHTI), Orléans, France. [7]Center for High Pressure Science and Technology Advanced Research, Beijing, China. ✉e-mail: feihongzhan@zju.edu.cn

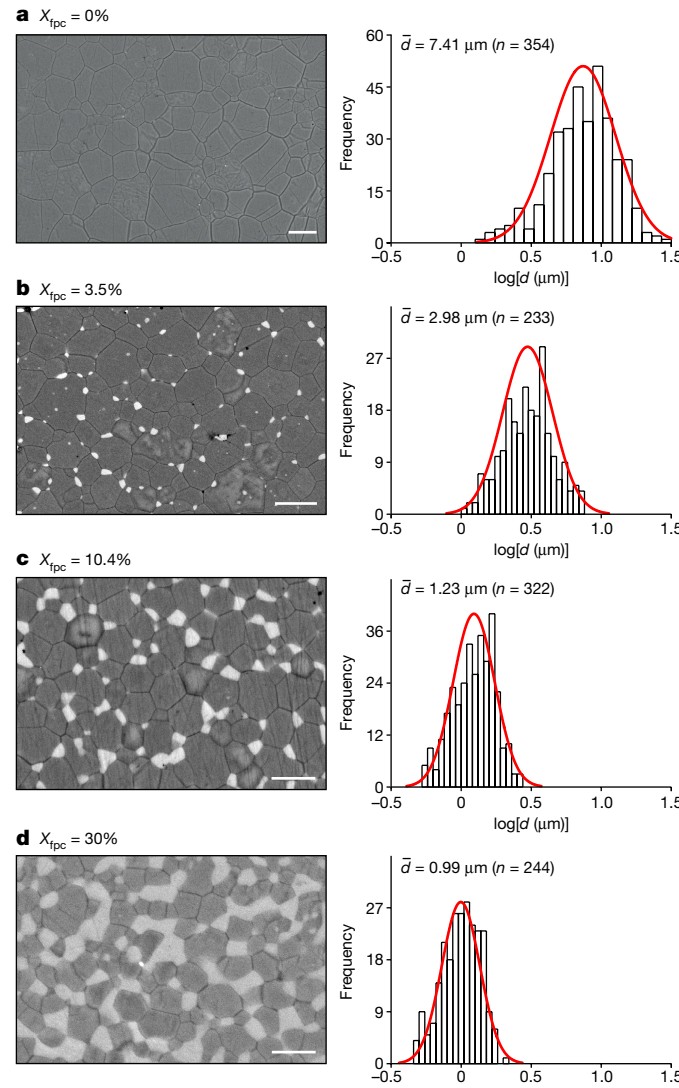

**a** $X_{fpc} = 0\%$

$\bar{d} = 7.41\ \mu m\ (n = 354)$

**b** $X_{fpc} = 3.5\%$

$\bar{d} = 2.98\ \mu m\ (n = 233)$

**c** $X_{fpc} = 10.4\%$

$\bar{d} = 1.23\ \mu m\ (n = 322)$

**d** $X_{fpc} = 30\%$

$\bar{d} = 0.99\ \mu m\ (n = 244)$

**Fig. 1 | Bridgmanite grain sizes after annealing at 27 GPa and 2,200 K for 100 min. a–d**, Backscattered electron images (dark, bridgmanite; bright, ferropericlase) and grain-size distribution. $n$, number of analysed grains; $\bar{d}$, average grain size obtained from mean log($d$), which decreases with increase in $X_{fpc}$. Scale bars, 10 μm (**a**), 5 μm (**b**) and 2 μm (**c** and **d**).

at a fixed $X_{fpc}$ of 30% (refs. 37,38). As the lower mantle consists of both pyrolitic rocks with high $X_{fpc}$ and bridgmanite-enriched rocks with low $X_{fpc}$ as discussed above[5–7,17–19], the influence of the proportion of ferropericlase on bridgmanite growth rate needs to be investigated.

Here we investigated the grain-growth kinetics of bridgmanite as a function of $X_{fpc}$ by multi-anvil high-pressure experiments. Aggregates of bridgmanite with different $X_{fpc}$ (about 0–60%) were pre-synthesized from San Carlos olivine, orthopyroxene (opx), solution–gelation-derived silicates (sol–gel) and melt-quenched silicate glasses (Extended Data Table 1) and annealed at 27 GPa and 2,200 K for 1.5–1,000 min for grain growth (Extended Data Table 2). The grain sizes were obtained from backscattered electron images of the recovered samples (Fig. 1), from which the growth-rate constant was calculated. Details of the experiment are provided in the Methods.

## Evolution of grain size over time

The recovered samples show that the grain-size distribution in log units (log($d$)) follows a Gaussian distribution (Fig. 1). As expected, the

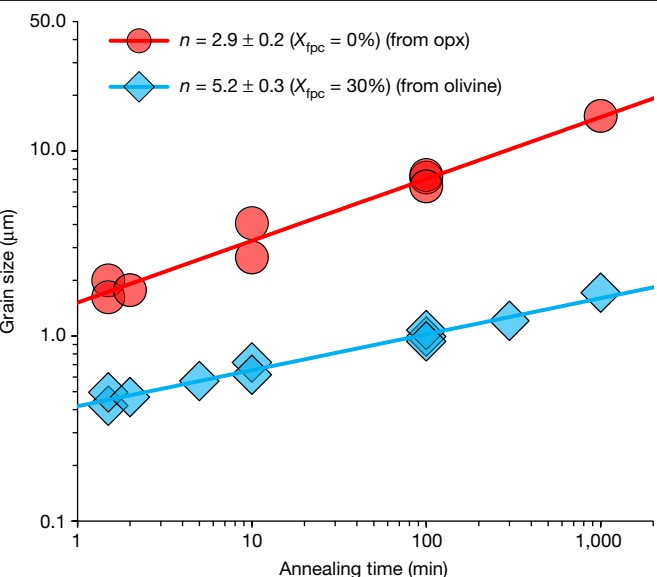

- ● $n = 2.9 \pm 0.2$ ($X_{fpc} = 0\%$) (from opx)
- ◆ $n = 5.2 \pm 0.3$ ($X_{fpc} = 30\%$) (from olivine)

**Fig. 2 | Evolution of bridgmanite grain size over time.** After annealing at 27 GPa and 2,200 K, for the indicated annealing time, the grain size of bridgmanite in the single-phase system ($X_{fpc} = 0\%$, from opx) is significantly larger than the aggregates with 30% of ferropericlase ($X_{fpc} = 30\%$, from olivine). The grain-size exponent $n$ is smaller when $X_{fpc} = 0\%$, indicating faster grain-size evolution over time.

mean grain size increases with an increase in annealing duration for both single-phase ($X_{fpc} = 0\%$) and two-phase aggregates (Fig. 2). After annealing at 2,200 K for 1.5–1,000 min, the grain size of samples with $X_{fpc} = 0\%$ is 0.7–1.0 orders of magnitude larger than those with $X_{fpc} = 30\%$ (Fig. 2). Samples pre-synthesized from different starting materials (olivine, opx, sol–gel and glasses) show consistent results (Fig. 3a–c).

Grain growth of polycrystalline aggregates follows a power law that can be approximated by

$$d^n - d_0^n = kt, \tag{1}$$

where $d$ denotes grain size after a growth experiment of duration $t$, $d_0$ is the initial grain size, $k$ is the growth-rate constant and $n$ is the grain-size exponent (coarsening exponent). For our annealing durations, $d$ exceeds $d_0$ by more than a factor of three (Extended Data Fig. 1); therefore, $d_0$ can be neglected in equation (1). Hence, log($d$) increases approximately linearly with increasing log($t$) (Fig. 2). The slopes of the fitting lines represent $1/n$ in equation (1).

Least-squares fitting of our data yields $n = 2.9 \pm 0.2$ and $5.2 \pm 0.3$ for $X_{fpc} = 0\%$ and 30%, respectively[37] (Fig. 2). These two values of $n$ agree well with those obtained from theoretical models—that is, $n = 2$–3 for grain growth controlled by grain-boundary diffusion in a single-phase system and $n = 4$–5 for a two-phase system[39,40], and are comparable to those reported for other minerals such as olivine, wadsleyite and ringwoodite (single phase)[41–43] as well as olivine–pyroxene and forsterite–nickel aggregates (two phases)[44,45]. For intermediate $X_{fpc}$, the grain size also increases with increasing duration (Fig. 3a–c). However, $n$ ranges from 3.1 to 6.2 because of the scatter of data points (Fig. 3d).

## Effects of $X_{fpc}$ on the rate of grain growth

The growth rate of bridgmanite is found to be significantly reduced by the presence of ferropericlase. After annealing for 1.5–100 min, the grain size of samples with $X_{fpc} \approx 10\%$ is smaller by 0.5–0.8 orders of magnitude than for $X_{fpc} = 0\%$, but at higher $X_{fpc}$ (up to about 60%) the ferropericlase proportion has a minor effect (Fig. 3a–c). This decrease in grain size with increasing $X_{fpc}$ cannot be ascribed to differences in

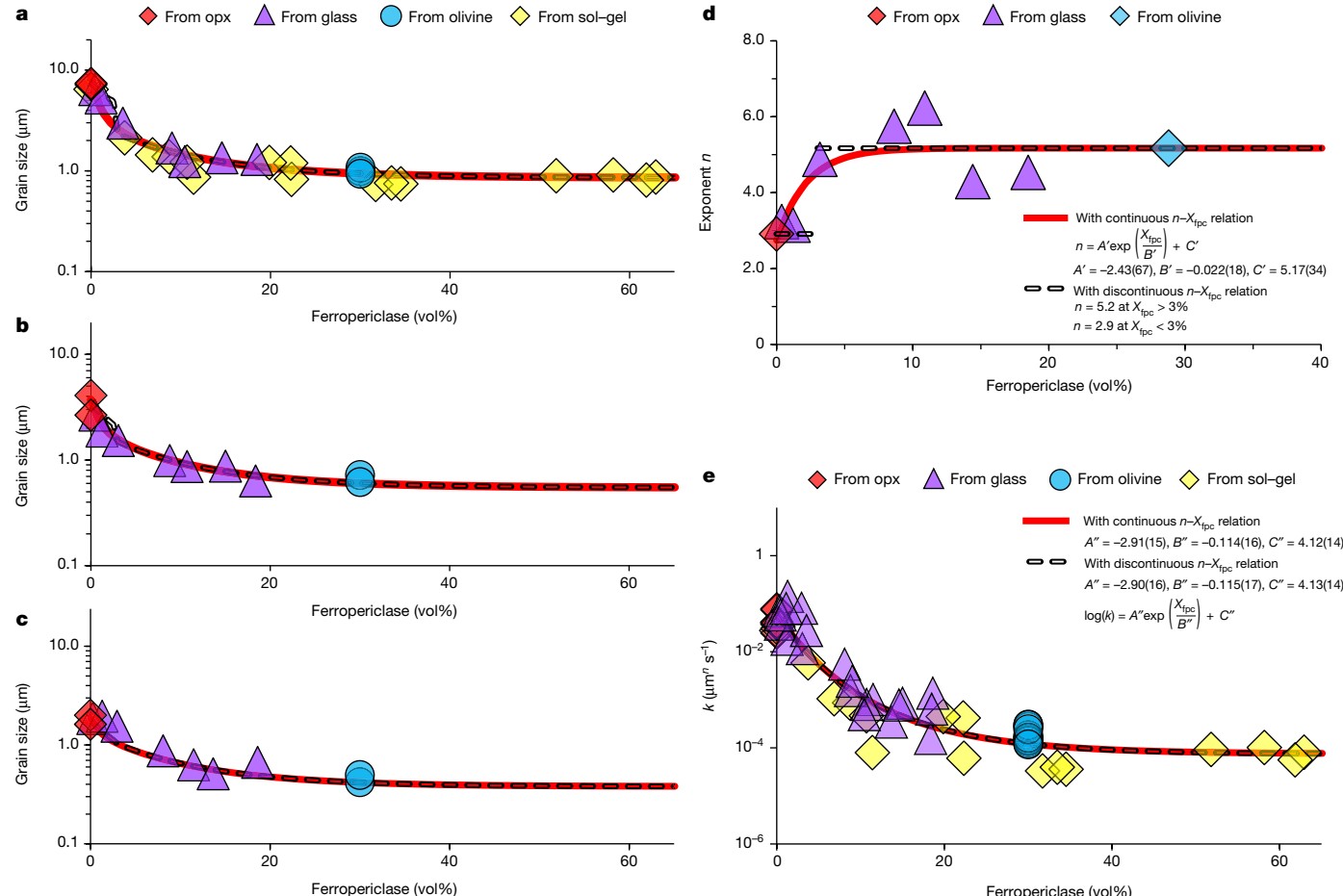

**Fig. 3 | Grain size of bridgmanite, grain-size exponent and growth-rate constant as a function of $X_{fpc}$. a–c**, Grain sizes after annealing at 27 GPa, 2,200 K for 100 min (**a**), 10 min (**b**) and 1.5 min (**c**). Samples synthesized from different starting materials (olivine, opx, sol–gel and glasses) show consistent results. **d**, Grain-size exponent $n$. **e**, Growth-rate constant $k$. The solid line in **d** is obtained by assuming that $n$ increases continuously with an increase in $X_{fpc}$ following the empirical equation $n = A'\exp(X_{fpc}/B') + C'$, whereas the dashed line represents a discontinuous change of $n$ with $X_{fpc}$—that is, $n = 2.9$ at $X_{fpc} < 3\%$ and $n = 5.2$ at $X_{fpc} > 3\%$. Accordingly, the solid and dashed lines in **e** are fitting curves of $k$ to the equation $\log(k) = A''\exp(X_{fpc}/B'') + C''$ ($k$ in units of $\mu m^n s^{-1}$) based on the continuous and discontinuous $n$, respectively. The fitting parameters are shown in the figure. The solid and dashed lines in **a–c** are calculated from the $n–X_{fpc}$ and $k–X_{fpc}$ relations in **d** and **e**.

Fe content for two reasons. First, our samples did not show a large variation in Fe contents (Extended Data Table 3). Second, bridgmanite synthesized from olivine (Fe/(Mg + Fe) ≈ 10%) and from Fe-free forsterite show only a difference in grain size of 0.1 log units[37].

As shown above, although the exponents $n$ for $X_{fpc} = 0\%$ and 30% are well constrained (Fig. 2), the $n–X_{fpc}$ relation is unknown because of the scatter of data points for intermediate $X_{fpc}$ (Fig. 3d). The exponent $n$ may change with $X_{fpc}$ either continuously or discontinuously. We therefore fit the data points to both continuous and discontinuous $n–X_{fpc}$ models in Fig. 3d. In either case, the growth-rate constant $k = d^n/t$ ($k$ in units of $\mu m^n s^{-1}$) decreases with increasing $X_{fpc}$. The fitting curves of $k–X_{fpc}$ based on the two $n–X_{fpc}$ models are essentially the same (Fig. 3e).

Grain growth in a two-phase system is controlled by growth of the matrix (bridgmanite) and coarsening of the second phase (ferropericlase) by Ostwald ripening. If ferropericlase coarsening does not occur, the grain size of bridgmanite should be limited by a constant value of the interparticle spacing of ferropericlase ($\bar{r}$, the average distance between adjacent ferropericlase grains). To understand whether ferropericlase coarsening occurs or not, the changes in $\bar{r}$ and $d_{fpc}$ (grain size of ferropericlase) over time are examined. It is found that $d_{fpc}$ increases with time in both low-$X_{fpc}$ (approximately 3–3.5%) and high-$X_{fpc}$ (approximately 18.5%) samples with similar rates as bridgmanite, whereas $\bar{r}$ increases with time systematically and is linearly propor-

tional to the grain size of bridgmanite (Extended Data Figs. 2 and 3). Therefore, both $d_{fpc}$ and $\bar{r}$ indicate simultaneous ferropericlase coarsening and bridgmanite growth. The growth rate of bridgmanite is affected by ferropericlase even at low $X_{fpc}$ (for example, about 3%) (Fig. 3), which is characteristic of two-phase systems in general[44,46].

## Variation in viscosity with $X_{fpc}$

Our experimental results indicate that the grain growth rate of bridgmanite-enriched rocks should be much faster (two to three orders of magnitude larger in $k$ as shown in Fig. 3d) than that of pyrolitic rocks. The growth-rate contrast should readily cause a grain-size contrast and this grain-size contrast increases further with geological time (Fig. 2). Over a short timescale of 10 Myr (that is, shortly after magma ocean crystallization) at a temperature of 2,200 K (typical mid-mantle temperatures[47]), the grain size of bridgmanite-enriched rocks already exceeds that of pyrolitic rocks by about two orders of magnitude. Over a timescale of 4.5 Gyr (that is, the whole history of Earth), the grain-size difference reaches around 2.5 orders of magnitude (Fig. 4a).

To infer the viscosity contrast of rocks with variable $X_{fpc}$, the diffusion- and dislocation-creep rates are calculated as a function of $X_{fpc}$ based on the growth rate of bridgmanite determined in this study and the Si diffusivity determined in previous studies given in Extended Data Table 4

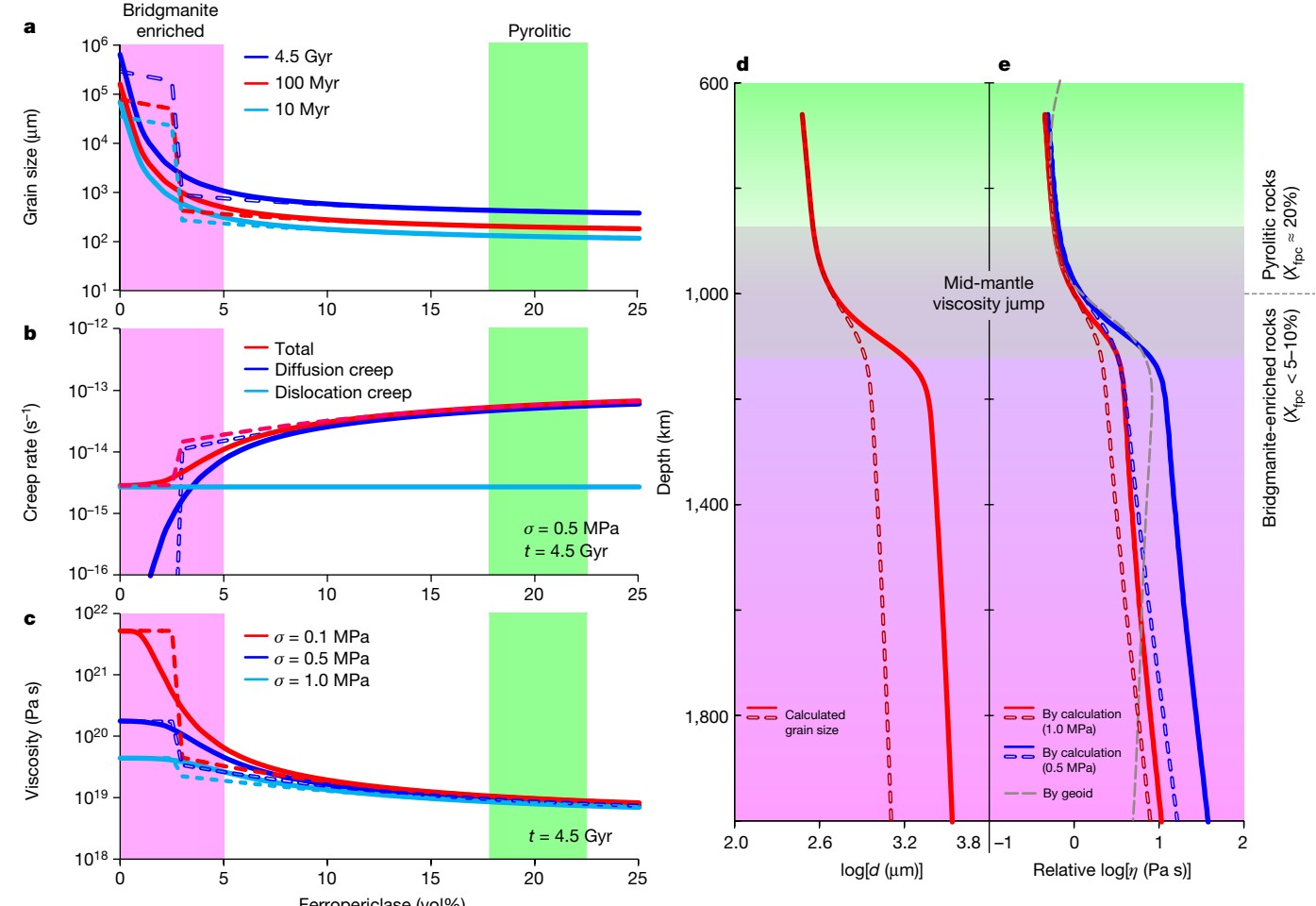

**Fig. 4 | Variation in grain size, creep rate and relative viscosity with $X_{fpc}$ and with depth in the lower mantle. a**, Grain size of bridgmanite calculated for growth over geological timescales of 10 Myr to 4.5 Gyr at 2,200 K. **b**, Simulated creep rates at 2,200 K assuming a stress of 0.5 MPa and grain size after growth for 4.5 Gyr. **c**, Relative viscosity at 2,200 K at stresses of 0.1–1 MPa (where $\sigma$ denotes stress) and grain size after 4.5 Gyr. **d**, Variation in grain size with depth along a lower-mantle geotherm[47] after 4.5 Gyr by assuming $X_{fpc}$ = 5% in

bridgmanite-enriched rocks and $X_{fpc}$ = 20% in pyrolitic rocks. **e**, Comparison of relative viscosity based on geophysical observations[1] (thick grey curve) and calculations with grain size from **d** at a stress of 1.0 MPa (red curves) and 0.5 MPa (blue curves). The solid and dashed lines represent calculations based on the continuous and discontinuous variations in $n$ with $X_{fpc}$ given in Fig. 3d, respectively. Note that the viscosity profiles in the figure represent only the relative changes with depth.

(for calculation details and uncertainty analysis, see Methods and Extended Data Figs. 4 and 5). Because of the inverse power relation, a grain-size contrast of two orders of magnitude causes the diffusion-creep rate of pyrolitic rocks that is more than four orders of magnitude higher than that of bridgmanite-enriched rocks (Fig. 4b). By contrast, the dislocation-creep rate is independent of grain size. As a result, the total creep rate of pyrolitic rocks remains one to two orders of magnitude higher (Fig. 4b) and, therefore, the viscosity is accordingly lower than that of bridgmanite-enriched rocks (Fig. 4c). Although the magnitude of the viscosity contrast depends on the stress conditions because of the contribution of dislocation creep (Fig. 4c), the non-hydrostatic stress in most of the mantle of Earth is estimated to be ≤1.0 MPa (ref. 48) or even ≤0.3 MPa (ref. 24). In this case, the grain-size contrast always causes a significant viscosity contrast even if dislocation creep dominates in the bridgmanite-enriched rocks (Fig. 4c and Methods).

## Viscosity jump in the mid-mantle

Our results provide an explanation for the long-term preservation of bridgmanite-enriched rocks in the deep lower mantle as indicated by

geophysical and geodynamical constraints[5–7,18–20]. Bridgmanite-enriched rocks formed in the deep lower mantle at the early stage of the history of Earth because of magma ocean crystallization[13–15] are expected to have developed grain sizes that are more than two orders of magnitude larger, and therefore have a much higher viscosity, than the overlying pyrolitic rocks in around 100 Myr or less (Fig. 4a,c). The high viscosity of these early-developed bridgmanite-enriched rocks should prevent them from being mixed with pyrolitic rocks over the age of Earth, leading to their preservation over geological timescales[5–7]. By contrast, pyrolitic rocks are gravitationally stable at the topmost and bottom layers of the lower mantle[20]. Therefore, these rocks may circulate around the bridgmanite-enriched rocks through narrow and rheologically weak channels[5,7].

The mid-mantle viscosity jump[1] can thus be explained by the grain-size contrast between bridgmanite-enriched rocks and the overlying pyrolite. Along a typical geotherm[47], the grain size of bridgmanite in each rock continuously increases with depth as temperature increases, and a grain-size increase of about one order of magnitude (based on the continuous $n$ in Fig. 3d) occurs at 800–1,200-km depth owing to the transition from pyrolitic-to-bridgmanite-enriched rocks with depth (Fig. 4d). Accordingly, a viscosity increase by about one

order of magnitude is sustained (for a stress of 1.0 MPa), which agrees with the geophysically constrained viscosity jump in the mid-mantle (Fig. 4e). For lower stresses, the viscosity increase would be even larger—that is, about 1.3 orders of magnitude for a stress of 0.5 MPa (Fig. 4e). Although the viscosity increase at 800–1,200-km depth is smaller using the discontinuous $n$ model (Fig. 3d), it is still about one order of magnitude for a stress of about 0.5 MPa (Fig. 4e). Note that the experimental pressure conditions in this study were limited to 27 GPa, corresponding to a depth of 800 km. Considering a negative pressure dependence of grain growth[43], the grain size as well as the viscosity of pyrolitic rocks decreases with depth. By contrast, the viscosity of bridgmanite-enriched rocks is independent of grain size because of the dominance of dislocation creep. Thus, the viscosity contrast between pyrolitic and bridgmanite-enriched rocks is expected to be even larger.

Our main finding that the grain-growth rate increases sharply with bridgmanite enrichment thus provides a unified explanation for the preservation of ancient bridgmanite-enriched rocks over geological timescales[5,7] and the present-day viscosity jump in the mid-mantle[1] (Fig. 4e). Although the grain-size increase with depth may not occur globally at 800–1,200-km depth, it should be sufficient to affect a wide range of geophysical and geochemical processes. For example, the sinking of slabs may be slowed down in the regions in which they encounter high-viscosity bridgmanite-enriched rocks, leading to slab stagnation at about 1,000-km depth as indicated by seismic observations[3]. The plumes ascend vertically through the bridgmanite-enriched deep lower mantle[4], but they may be deflected at about 1,000-km depth because of the horizontal flow promoted in the pyrolitic rocks just above the viscosity jump as shown by full-waveform seismic tomography[4]. Furthermore, the bridgmanite-enriched rocks may sustain widespread seismic reflectors[49], host primordial geochemical anomalies (for example, $^{142}$Nd, $^{182}$W and $^{3}$He) in the deep mantle[9,10] and balance the discrepancy in Mg:Si ratio between upper-mantle rocks (Mg:Si ≈ 1.3) and the building blocks of Earth[8] (chondrites, Mg:Si ≈ 1.05).

The lower-mantle rheological structure as predicted by our grain-size model may further explain the lack of observed seismic anisotropy. In the pyrolitic shallow lower mantle, diffusion creep dominates because of the small grain sizes (Fig. 4b), leading to the absence of seismic anisotropy[11]. In turn, because of the high viscosity, the bridgmanite-enriched deep lower mantle may accumulate little strain and thus no anisotropy owing to the high viscosity[5–7], despite the dominance of dislocation creep (Fig. 4b). Anisotropy in the lower mantle is therefore restricted to regions with high stress and significantly accumulated strains such as near subducting slabs, leading to locally enhanced seismic anisotropy[12].

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

## Methods

### Starting materials

Four types of starting material were used in this study: (1) olivine powder with a composition of $(Mg,Fe)_2SiO_4$; (2) opx powder with a composition of $(Mg,Fe)SiO_3$; (3) sol–gel-derived silicate powders with bulk compositions of $(Mg,Fe)_{1.5}SiO_{3.5}$, $(Mg,Fe)_{1.25}SiO_{3.25}$ and $(Mg,Fe)_{1.125}SiO_{3.125}$; (4) silicate glass powders with bulk compositions of $(Mg,Fe)_xSiO_{2+x}$ ($x = 1.5, 1.4, 1.3, 1.2, 1.1, 1.05$ and $1.02$). The Mg:Fe atomic ratios in all of the powders were about 9:1.

Material 1 was prepared by grinding hand-picked single crystals of San Carlos olivine. Material 2 was prepared from MgO, FeO and $SiO_2$ oxides. Both materials 1 and 2 were used in a previous study[50]. Material 3 was prepared from tetraethyl orthosilicate and metallic Mg and Fe dissolved in dilute nitric acid following the procedure reported in ref. 51. The powders have compositions between those of materials 1 and 2 to trace the grain-growth kinetics as a function of $X_{fpc}$. However, the products of material 3 after high-pressure synthesis were found to have inhomogeneous ferropericlase distributions as described in the section below (Extended Data Fig. 1). Therefore, the silicate glasses (material 4) were prepared by quenching the oxide melts with $(Mg,Fe)_xSiO_{2+x}$ bulk compositions ($x$ as described above) from about 2,500 K (estimated with an optical pyrometer) to room temperature in an aerodynamic levitator equipped with a two-$CO_2$-laser heating system[52]. The products of material 4 after high-pressure synthesis have uniform ferropericlase distributions (Extended Data Fig. 1). The powders of materials 3 and 4 were annealed at 1,100 K for 24 h in an ambient-pressure $CO–CO_2$ gas-mixing furnace with oxygen partial pressure controlled at approximately 0.5 log units above the iron–wüstite buffer to reduce the ferric iron to a ferrous state. All the powders were stored in a vacuum furnace at 400 K before use.

### Synthesis of bridgmanite–ferropericlase aggregates

Bridgmanite with various fractions of ferropericlase was synthesized from the above-mentioned starting materials using a multi-anvil press. The detailed synthesis procedures have been described previously[37]. In brief, multiple layers of starting materials separated by Fe foils were loaded into Pt capsules with outer and inner diameters of 1.0 and 0.8 mm, respectively. The thickness of each layer was about 0.15 mm. Small amounts of Fe–FeO powder were loaded next to the Fe foils to buffer the oxygen fugacity. High-pressure experiments were performed by the multi-anvil technique using a $Cr_2O_3$-doped MgO octahedral pressure medium with a 7-mm edge length with a $LaCrO_3$ furnace and tungsten carbide anvils with a 3-mm truncation edge length (7/3 assembly). The pressure and temperature conditions were 27 GPa and 1,700 K, respectively. The heating duration was 5 min. The run conditions and products are summarized in Extended Data Table 1.

Homogeneously distributed bridgmanite–ferropericlase mixtures with a grain size much less than 0.1 μm (post-spinel) and single-phase bridgmanite with a grain size of approximately 0.42 μm (opx–bridgmanite) were synthesized from materials 1 and 2, respectively (Extended Data Fig. 1a,b). The samples synthesized from material 3 have an inhomogeneous distribution (locally homogeneous) of bridgmanite and ferropericlase grains (Extended Data Fig. 1c), probably because of an inhomogeneous Si distribution during gelation. The grain size is approximately 0.15 μm. The samples synthesized from material 4 seemed to be homogenous, with a grain size of about 0.2 μm (Extended Data Fig. 1d).

### Grain-growth experiments

All the synthesized aggregates were mechanically broken into small pieces (each 100–200 μm in size). Multiple pieces were embedded in pre-dried CsCl powder in Pt capsules, which provided quasi-hydrostatic conditions[50,53]. An Fe–FeO powder was loaded at the two ends of the Pt capsules to buffer the oxygen fugacity ($f_{O_2}$). The capsules were loaded into the 7/3 multi-anvil cell assemblies and compressed to 27 GPa, followed by heating at 2,200 K for 1.5–1,000 min (Extended Data Table 2). Because of the relatively fast heating and cooling speeds (2–3 min for heating from 1,700 to 2,200 K and less than 1 s for cooling from 2,200 K to below 1,700 K), the growth during heating and cooling is negligible.

### Sample analysis

The recovered samples were separated from CsCl by dissolution in water, polished and observed using a scanning electron microscope with acceleration voltages of 5–20 kV. Bridgmanite and ferropericlase grains were distinguished by the brightness contrast in backscattered electron (BSE) images (Fig. 1). The volume fraction of ferropericlase was obtained from the BSE images. The area of each bridgmanite grain was determined using an image processing software (ImageJ). The grain size ($d$) of each grain was obtained from the diameter of the area-equivalent circle. The grain size in log units ($\log(d)$) showed a Gaussian distribution (Fig. 1); therefore, the mean grain sizes ($\bar{d}$) were calculated from the mean $\log(d)$ based on the Gaussian distribution[37].

The bridgmanite and ferropericlase grains were homogeneously distributed in the post-spinel, opx-bridgmanite and glass samples. More than 130 bridgmanite grains were analysed for each sample (Extended Data Table 2). In the sol–gel samples, BSE images were taken on locally homogenous areas. Each data point of the sol–gel samples (Fig. 3a) represents the grain size and $X_{fpc}$ in an individual BSE image. As mentioned above, the heterogeneity had occurred during the sample synthesis procedure, after which the grains already reached an equilibrated texture (120° triple junction, Extended Data Fig. 1c). Therefore, the grain growth in each locally homogenous area during the annealing experiment should not be affected. This is confirmed by the consistent results obtained in the sol–gel, glass, opx-bridgmanite and post-spinel samples. Some metallic iron particles that locally appeared in the sol–gel samples (Supplementary Figs. 49–53) are also expected to have a negligible effect on the $\log(d)$–$X_{fpc}$ relation because of its small volume fraction in comparison with ferropericlase.

The mean interparticle spacing ($\bar{r} = 1/\rho^{1/2}$) was calculated from the two-dimensional density of ferropericlase (where $\rho$ is the number of ferropericlase particles per μm$^2$). Note that $\bar{r}$ becomes invalid for $X_{fpc} = 0\%$ and becomes inappropriate for the high-$X_{fpc}$ samples (greater than about 30%) in which ferropericlase grains are significantly or completely interconnected (Extended Data Table 2).

The chemical compositions of bridgmanite after grain growth were analysed using an electron probe microanalyser (EPMA). An acceleration voltage of 15 kV and a beam current of 5 nA were used. The counting time was 20 s for each point analysis. An enstatite crystal and metallic iron were used as standards for Mg, Si and for Fe, respectively. The results of the EPMA analysis are listed in Extended Data Table 3.

### Calculation of creep rates and viscosity

**Flow laws of dislocation creep and diffusion creep.** The diffusion-creep ($\dot{\varepsilon}_{diff}$) and dislocation-creep ($\dot{\varepsilon}_{dis}$) rates are calculated using flow laws of Coble and Nabarro–Herring diffusion creep[54,55] and of pure-climb controlled dislocation creep[56,57], respectively, based on the grain size of bridgmanite determined in this study and Si diffusion coefficients from previous studies[58–60]:

$$\dot{\varepsilon}_{diff} = A\frac{\sigma V_m}{RTd^2}\left(D^{lat} + \frac{\delta D^{gb}}{d}\right) \tag{2}$$

$$\dot{\varepsilon}_{dis} = \frac{D^{lat}b\sigma^3 V_m}{\pi RTG^2}\ln\left(\frac{4G}{\pi\sigma}\right), \tag{3}$$

where $A$ is a constant ($A = 16/3$); $G$ is the shear modulus (about 210 GPa); $V_m$ is the molar volume (25.5 cm$^3$ mol$^{-1}$); $b$ is the Burgers vector (0.5 nm); $D^{lat}$ and $D^{gb}$ are the lattice and grain-boundary diffusion coefficients of

the slowest species (Si), respectively; $\delta$ is the grain boundary width; $\sigma$ is the stress; $R$ is the gas constant; and $T$ is the temperature[55]. The total creep rate is obtained by $\dot{\varepsilon}_{\text{total}} = \dot{\varepsilon}_{\text{diff}} + \dot{\varepsilon}_{\text{dis}}$, whereas $\eta$ is calculated from $\eta = \sigma/\dot{\varepsilon}_{\text{total}}$. The temperature dependences of $D^{\text{lat}}$ and $\delta D^{\text{gb}}$ in bridgmanite are taken from ref. 58 (the $D^{\text{lat}}$ obtained in refs. 59,60 is essentially the same as those of ref. 58, whereas the $\delta D^{\text{gb}}$ is systematically measured as only a function of temperature in ref. 58; detailed parameters are given in Extended Data Table 4). Their pressure dependences are unknown and are therefore assumed to be the same as those of olivine (1.7 and 4.0 cm³ mol⁻¹, respectively)[61,62].

**Uncertainty analysis.** Equations (2) and (3) are well-established principles for diffusion creep and dislocation creep, respectively, in ceramic materials and are commonly used to simulate the creep rates in minerals, especially for bridgmanite[24,56,57,59]. The validity of equation (3) is demonstrated by recent deformation experiments on bridgmanite in the dislocation-creep regime—that is, the dislocation-creep rate simulated by equation (3) is within uncertainty, which is consistent with those obtained in deformation experiments[24] (Extended Data Fig. 4a). Moreover, although deformation experiments on bridgmanite in the diffusion creep regime are impractical at present, the validity of equation (2) for diffusion creep is experimentally tested by other minerals such as olivine (figure 14 of ref. 63 and figure 9 of ref. 51) and pyroxene (Extended Data Fig. 4b).

Here we evaluate the uncertainty of the viscosity contrast between bridgmanite-enriched and pyrolitic rocks by the above calculations. The viscosity contrast is the ratio of creep rates between pyrolitic and bridgmanite-enriched rocks. Equations (2) and (3) suggest that the main uncertainties in the calculation come from the uncertainties of $D^{\text{lat}}$ and $\delta D^{\text{gb}}$. Because $D^{\text{lat}} \gg \delta D^{\text{gb}}/d$, in which $d \gg$ about 1 µm (ref. 58), both $\dot{\varepsilon}_{\text{diff}}$ and $\dot{\varepsilon}_{\text{dis}}$ become linearly proportional to $D^{\text{lat}}$ as shown in equations (2) and (3). The deformation of pyrolitic rocks is dominated by diffusion creep, whereas that of bridgmanite-enriched rocks is dominated by either diffusion or dislocation creep (depending on $X_{\text{fpc}}$ and $\sigma$) (Fig. 4b). If dislocation creep dominates in the bridgmanite-enriched rocks, the ratio of creep rates between pyrolitic and bridgmanite-enriched rocks becomes $\left(\frac{1}{d\sigma}\right)^2 \frac{\pi A G^2 \ln(4G/\pi\sigma)}{b}$. If diffusion creep dominates, the ratio is $(1/d)^2$. Therefore, in both cases the ratios of creep rates are independent of $D^{\text{lat}}$ and $\delta D^{\text{gb}}$. The uncertainties of $D^{\text{lat}}$ and $\delta D^{\text{gb}}$ (as well as their pressure and temperature dependences) thus affect only the absolute values of the simulated creep rate and viscosity, but do not affect the viscosity contrast between bridgmanite-enriched and pyrolitic rocks. As the uncertainties of the Burgers vector $b$ and shear modulus $G$ are negligible compared with the uncertainty of the viscosity contrast, the ratio of creep rates is only significantly controlled by $d$ and $\sigma$. The $\sigma$ in the general area of the mantle of Earth is small—that is, 0.1–1.0 MPa estimated from the velocities of upwelling and downwelling flows[48] and 0.02–0.3 MPa based on the deformation experiments of bridgmanite[24]. With $\sigma \leq 1.0$ MPa and $X_{\text{fpc}} \leq 5\%$ in bridgmanite-enriched rocks, the grain-size contrast always results in a viscosity contrast by more than one order of magnitude (Fig. 4c).

The pressure dependences of $D^{\text{lat}}$ and $\delta D^{\text{gb}}$, which are unknown, may affect the variation of $\eta$ with depth. Therefore, in addition to the calculations in Fig. 4e in which the activation volume for $D^{\text{lat}}$ ($\Delta V$) is assumed to be the same as that of olivine, $\eta$ is also calculated by assuming different $\Delta V$ values for $D^{\text{lat}}$ ($\Delta V$ for $\delta D^{\text{gb}}$ has a negligible effect because $D^{\text{lat}} \ll \delta D^{\text{gb}}/d$). As shown in Extended Data Fig. 5, $\Delta V$ affects the slope of *the* $\eta$–depth profile—that is, $\eta$ slightly decreases with increasing depth when $\Delta V$ is 0–1 cm³ mol⁻¹ and increases with depth when $\Delta V$ is 1–3 cm³ mol⁻¹. However, it does not affect the viscosity jump at around 1,000-km depth, which is reasonable because in the case of

either large or small $\Delta V$, $D^{\text{lat}}$ varies continuously with depth because the pressure and temperature increase continuously with depth. By contrast, $\Delta V > 3$ cm³ mol⁻¹ is unlikely because $\eta$ would increase by more than three orders of magnitude with depth from 660 to 2,000 km, which disagrees with the mantle viscosity profile estimated from geoid observations (Extended Data Fig. 5d).

## Data availability

The data of this manuscript are available at https://doi.org/10.5281/zenodo.7804779. Source data are provided with this paper.

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

**Acknowledgements** This work is supported by the Advanced Grant of the European Research Council (ERC) under the Horizon 2020 research and innovation programme of the European Union (no. 787527) and the research grant of Deutsche Forschungsgemeinschaft (DFG) (KA3434/19-1) to T.K., the annual budget of the Bayerisches Geoinstitut and the startup funding from Zhejiang University to H.F. and the NSF grant (NSF-EAR 2125895) to U.F. We appreciate A. Zandonà (CEMHTI) for introducing the levitator facility for making silicate glasses.

**Author contributions** H.F. designed the experiments, prepared the starting materials and performed high-pressure experiments, scanning electron microscope observations, EPMA analyses and data interpretation. W.C. and H.F. made the silicate glasses. H.F. initialized the geophysical implications and wrote the paper with comments from T.K., M.D.B., N.W. and U.F.; T.K. planned and organized this project; U.F. also planned the project independently and performed some preliminary experiments.

**Funding** Open access funding provided by Universität Bayreuth.

**Competing interests** The authors declare no competing interests.

**Additional information**
**Correspondence and requests for materials** should be addressed to Hongzhan Fei.

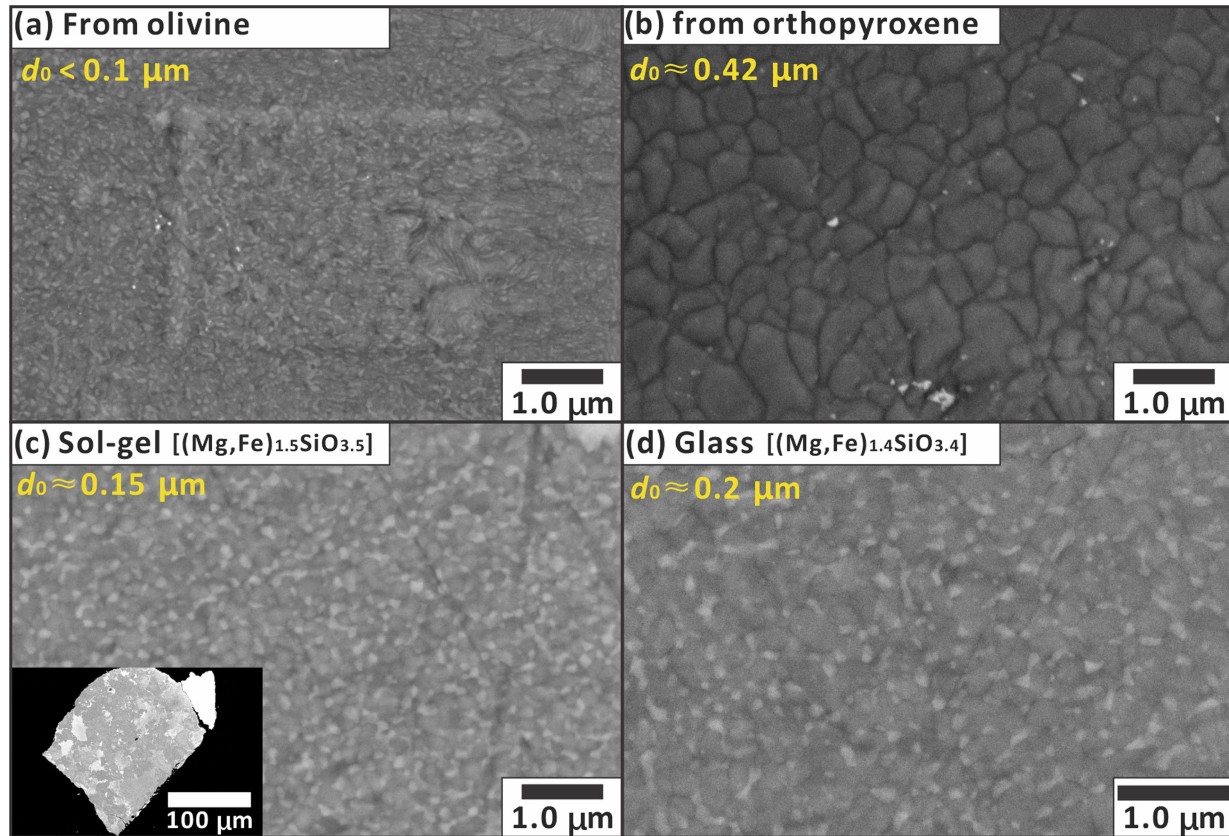

**(a) From olivine**
$d_0 < 0.1$ μm
1.0 μm

**(b) from orthopyroxene**
$d_0 \approx 0.42$ μm
1.0 μm

**(c) Sol-gel** [(Mg,Fe)$_{1.5}$SiO$_{3.5}$]
$d_0 \approx 0.15$ μm
100 μm
1.0 μm

**(d) Glass** [(Mg,Fe)$_{1.4}$SiO$_{3.4}$]
$d_0 \approx 0.2$ μm
1.0 μm

**Extended Data Fig. 1 | SEM images taken from the synthesized samples.**
(a) Post-spinel synthesized from olivine. The average grain size is <<0.1 μm. (b) Single phase of bridgmanite from orthopyroxene. The average grain size is 0.42 μm. (c) Bridgmanite + ferropericlase from sol-gel powder. The average grain size is approximately 0.15 μm. Insert: the whole view of this sample. The distribution of ferropericlase is inhomogeneous in the sol-gel sample. The sample consists of different domains with various $X_{fpc}$, but the $X_{fpc}$ in each domain is homogenous. (d) Bridgmanite + ferropericlase synthesized from glass with grain size of about 0.2 μm. Because of bridgmanite amorphization, it is not possible to take high-magnification and high-resolution images for (a), but the grain sizes are clearly much smaller (more than three times) than those after the grain growth runs (Extended Data Table 1) and $d_0$ is therefore negligible in Eq. (1).

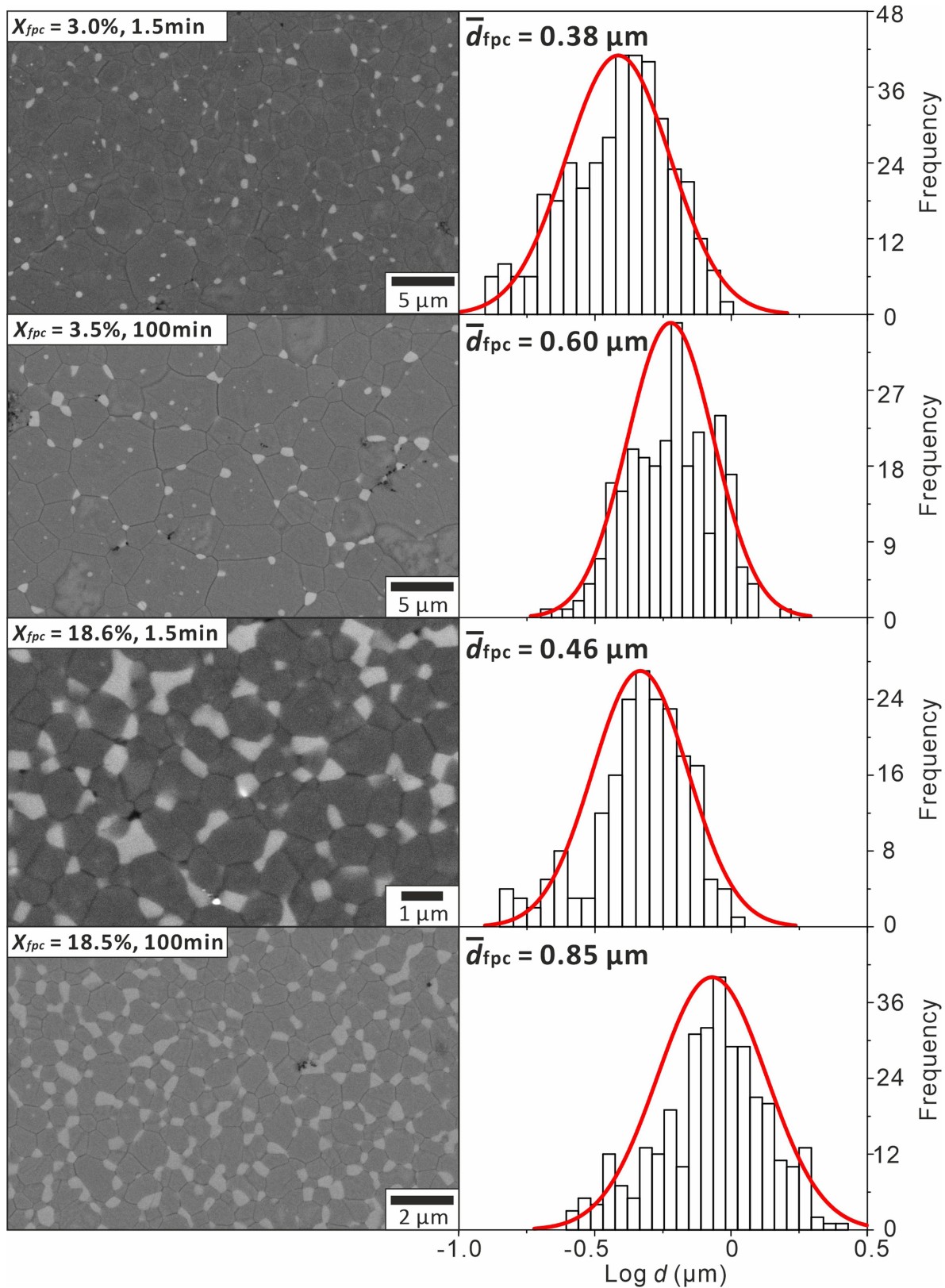

**Extended Data Fig. 2 | The grain size of ferropericlase increases with increasing experimental duration for both low $X_{fpc}$ (~ 3 – 3.5%) and relatively-high $X_{fpc}$ (~18.5%) samples.** With increasing duration from 1.5 to 100 min, the grain size of ferropericlase increases by a factor of ~1.6 (from 0.38 to 0.60 μm) and 1.8 (from 0.46 to 0.85 μm), respectively, for the $X_{fpc}$ = 3 – 3.5% and $X_{fpc}$ = ~18.5% samples. These rates are slightly lower but within error comparable with that of bridgmanite, which increases by a factor of ~1.9 (from 1.55 to 2.98 μm and from 0.67 to 1.30 μm as given in Extended Data Table 2).

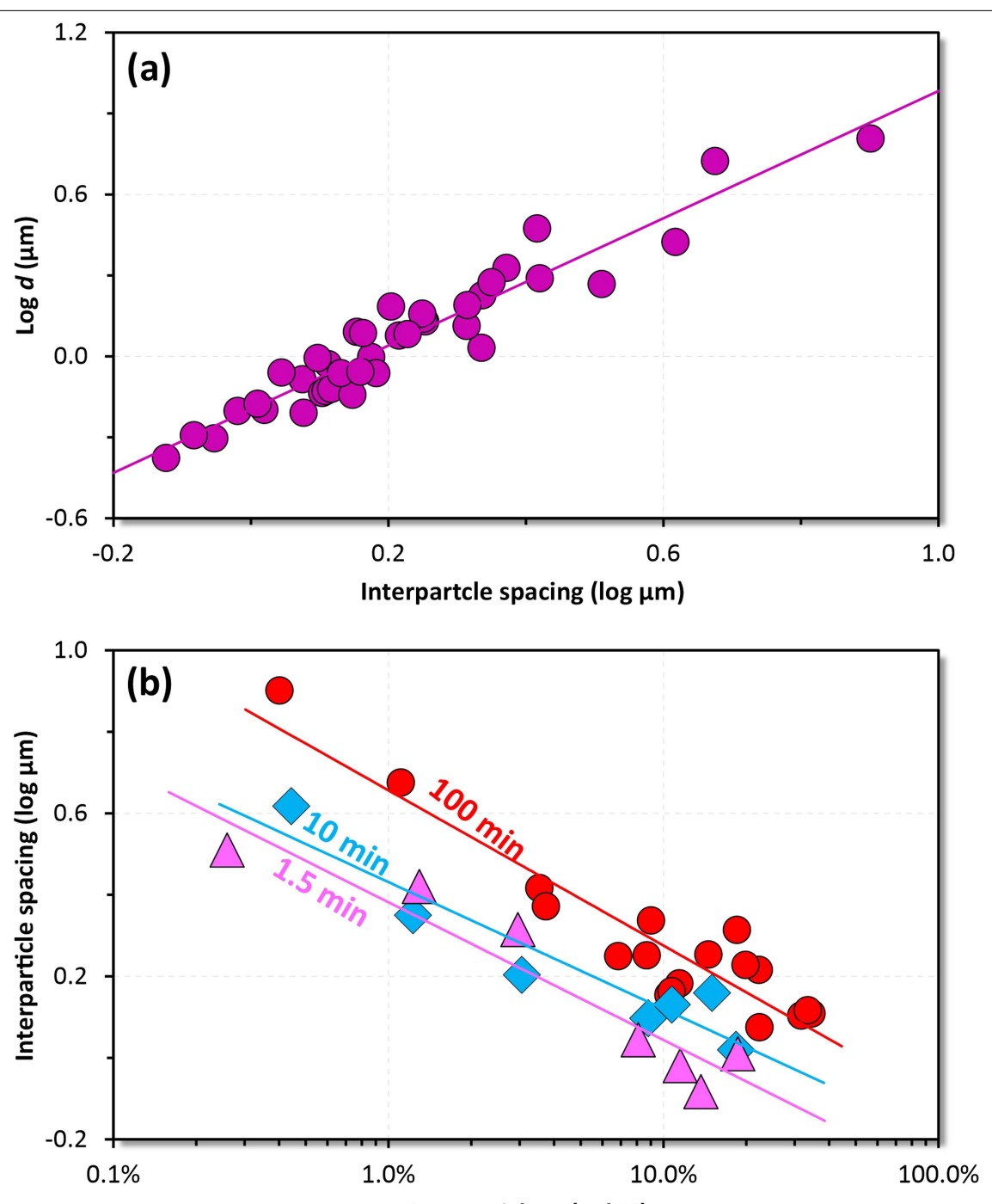

**Extended Data Fig. 3 | The grain size of bridgmanite (log *d*), interparticle spacing of ferropericlase (*r̄*), and ferropericlase volume fraction (*X_{fpc}*).** (a) The log *d* increases with increasing *r̄*. (b) the *r̄* is inversely proportional to *X_{fpc}*. The increase of *r̄* with duration indicates the coarsening of ferropericlase, while the correlation between log *d* and *r̄* indicates the simultaneous Ostwald ripening of ferropericlase and growth of bridgmanite. All the data with meaningful *r̄* from Extended Data Table 2 are plotted.

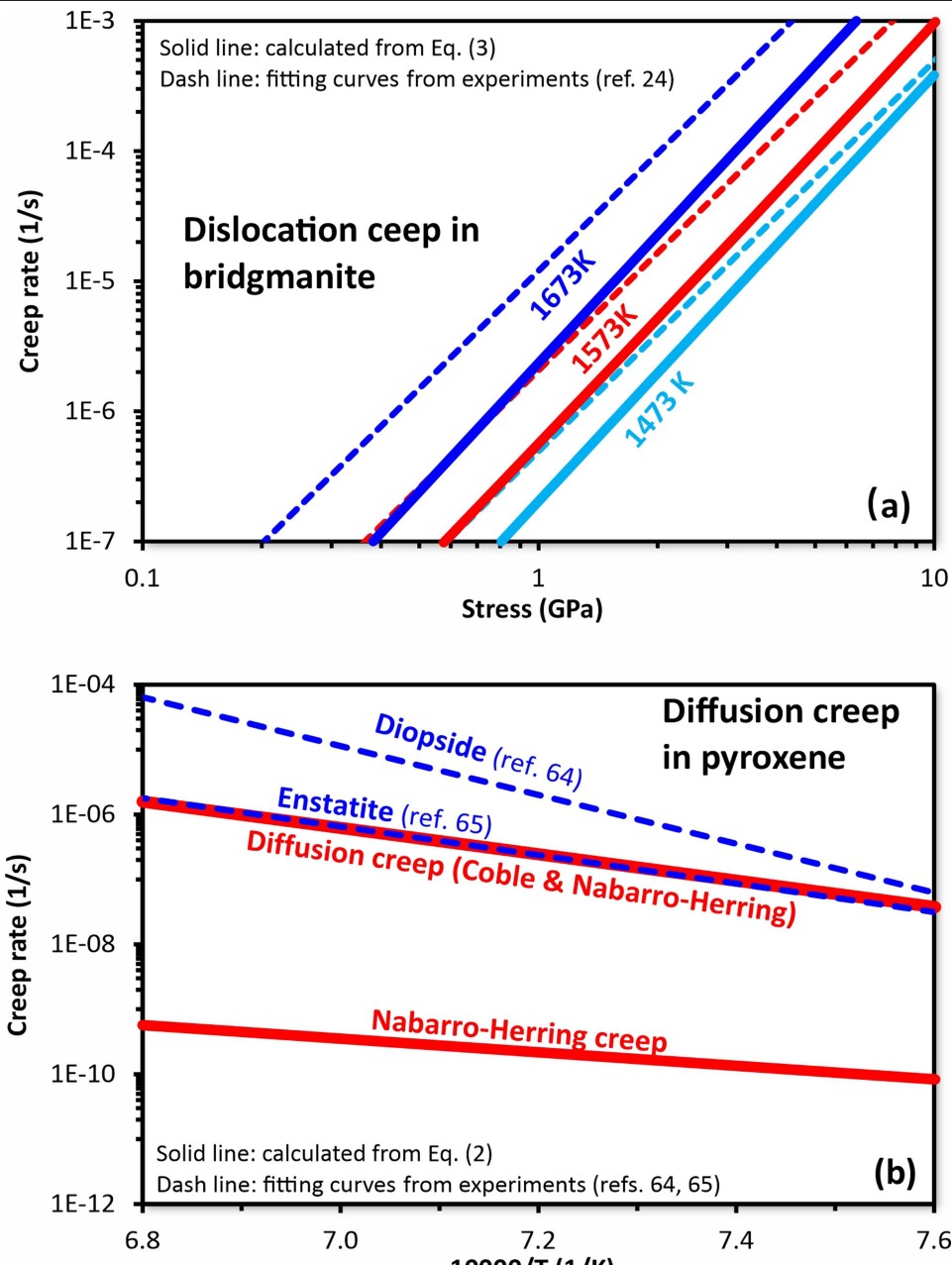

**Extended Data Fig. 4 | Consistency of creep rates simulated by Eqs. (2, 3) (solid lines) and measured by deformation experiments (dash lines).**
(a) Dislocation creep rate in bridgmanite from Eq. (3) and from experiments by Tsujino et al.[24] (b) Diffusion creep in pyroxene from Eq. (2) and from experiments by Ghosh et al. (diopside)[64] and Tasaka et al. (enstatite)[65] adjusted to a stress of 30 MPa and grain size of 1 μm (equivalent to the experimental conditions in refs. 64,65). The Si diffusion data for the calculations are from Xu et al. ($D_{Si}^{lat}$ in bridgmanite)[59], Fisler et al. ($D_{Si}^{gb}$ in enstatite)[66], and Bejina and Jaoul ($D_{Si}^{lat}$ in diopside)[67]. Note that Ghosh et al. [64] concluded the inconsistency of diffusion creep rates between calculations and experiments in diopside, however, they did not consider the Coble creep regime, which is not negligible in their small-grain-size samples (~1 μm). Additionally, the consistency in olivine is already demonstrated previously (Fig. 14 in ref. 63. and Fig. 9 in ref. 51).

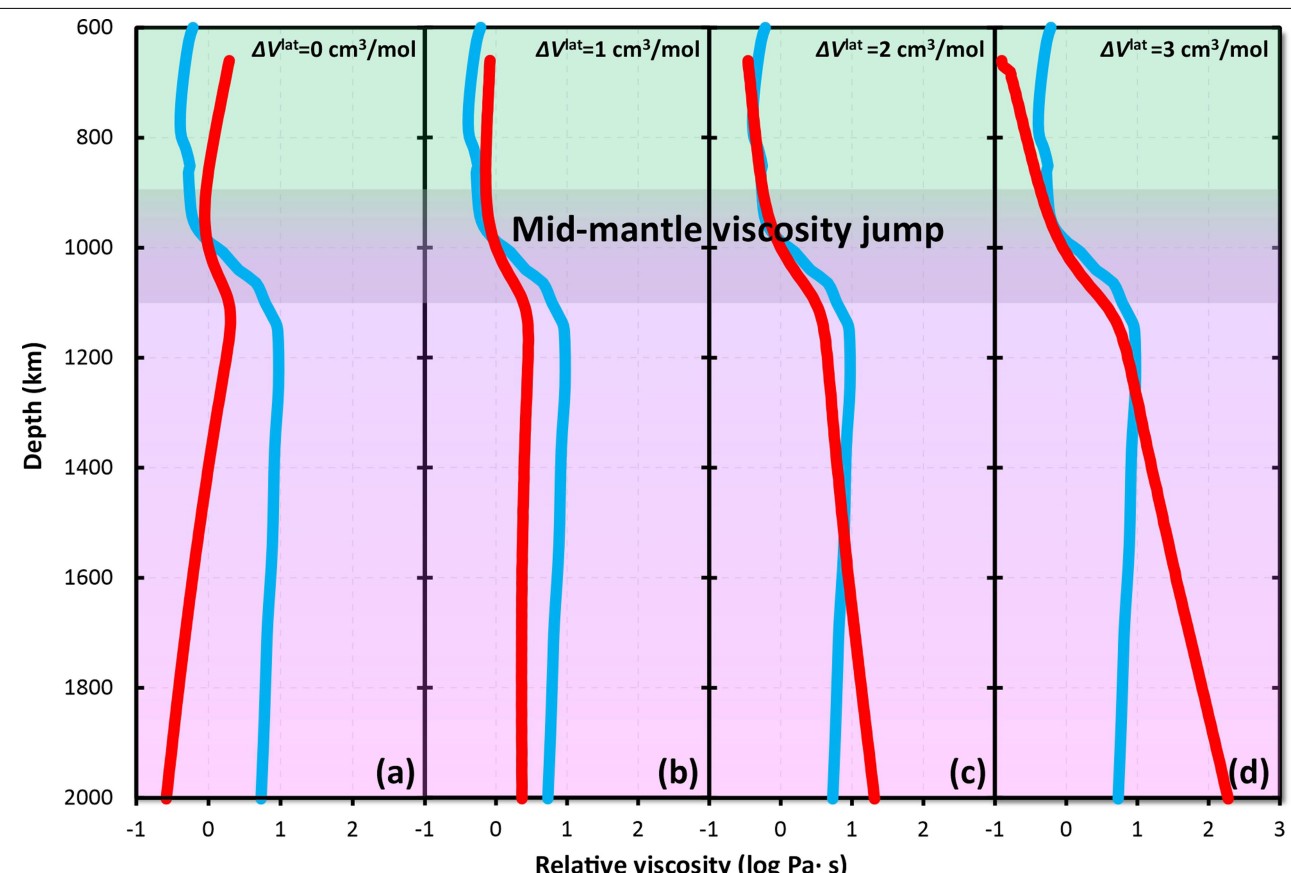

**Extended Data Fig. 5 | Calculated relative viscosity as a function of depth in the lower mantle by assuming different activation volumes ($\Delta V$ = 0 – 3 cm³/mol) for $D_\text{si}^\text{lat}$.** The red and blue curves represent the viscosity from calculations and from geoid analysis[1], respectively. The calculated viscosity is based on Eq. 2 and Eq. 3 with grain size variation with depth along a lower-mantle geotherm[47] after 4.5 Gyr and stress condition of 1.0 MPa. Note that the viscosity profiles in the figure only represent the relative changes of viscosity with depth, but not the absolute value of viscosity in the lower mantle.

**Extended Data Table 1 | Sample synthesis runs of high-pressure phase assemblages**

| Run No. | Starting material | Bulk composition | Sample name | Run products | $d_0$ (µm) |
|---------|-------------------|------------------|-------------|--------------|-----------|
| I926 | Olivine (i) | $(Mg,Fe)_2SiO_4$ | Post-spinel | Bridgmanite + ferropericlase ($X_{fpc} \approx 30\%$) (homogeneous ferropericlase distribution) | <0.1 |
|  | Orthopyroxene (ii) | $(Mg,Fe)SiO_3$ | Opx-bridgmanite | Bridgmanite ($X_{fpc} = 0\%$) | 0.42 |
| I1020 | Sol-gel (iii) | $(Mg,Fe)_{1.5}SiO_{3.5}$ | Sol-gel-1.5 | Bridgmanite + ferropericlase (inhomogeneous ferropericlase distribution, but locally homogenous) | ~0.15 |
|  |  | $(Mg,Fe)_{1.25}SiO_{3.25}$ | Sol-gel-1.25 |  |  |
|  |  | $(Mg,Fe)_{1.125}SiO_{3.125}$ | Sol-gel-1.125 |  |  |
| I1244 | Silicate glass (iv) | $(Mg,Fe)_{1.05}SiO_{3.05}$ | Glass-1.05 | Bridgmanite + ferropericlase (homogeneous ferropericlase distribution) | ~0.2 |
|  |  | $(Mg,Fe)_{1.1}SiO_{3.1}$ | Glass-1.1 |  |  |
|  |  | $(Mg,Fe)_{1.2}SiO_{3.2}$ | Glass-1.2 |  |  |
|  |  | $(Mg,Fe)_{1.4}SiO_{3.4}$ | Glass-1.4 |  |  |
| I1250 | Silicate glass (iv) | $(Mg,Fe)_{1.02}SiO_{3.02}$ | Glass-1.02 |  |  |
|  |  | $(Mg,Fe)_{1.3}SiO_{3.3}$ | Glass-1.3 |  |  |
|  |  | $(Mg,Fe)_{1.5}SiO_{3.5}$ | Glass-1.5 |  |  |

All runs were performed at a pressure of 27 GPa, temperature of 1700 K, with a duration of 5 min.

**Extended Data Table 2 | Run conditions, $X_{fpc}$, grain sizes ($d$) of bridgmanite in the run products, growth rates ($k$), and interparticle spacing ($\bar{r}$)**

| Run No. | $t$ (min) | Sample | Analyzed images | Analyzed grains | $X_{fpc}$ | Log$d$ (μm) | $\bar{d}$ (μm) | $k$ (μm$^n$/s)[*b] | $k$ (μm$^n$/s)[*c] | $\bar{r}$ (μm) |
|---|---|---|---|---|---|---|---|---|---|---|
| I930R | 1000 | Post-spinel[*a] | 5 | 756 | ~30% | 0.23 (15) | 1.72 | $2.72\times10^{-4}$ | $2.72\times10^{-4}$ | 2.88 |
| | | Opx-bridgmanite | 1 | 130 | 0% | 1.19 (15) | 15.5 | $3.06\times10^{-2}$ | $4.81\times10^{-2}$ | --[*d] |
| I951R | 10 | Post-spinel[*a] | 3 | 843 | ~30% | −0.14 (15) | 0.72 | $3.06\times10^{-4}$ | $3.06\times10^{-4}$ | 1.40[*d] |
| | | Opx-bridgmanite | 2 | 248 | 0% | 0.61 (26) | 4.08 | $7.88\times10^{-2}$ | $9.94\times10^{-2}$ | --[*d] |
| I928R | 1.5 | Post-spinel[*a] | 3 | 883 | ~30% | −0.38 (14) | 0.42 | $1.25\times10^{-4}$ | $1.25\times10^{-4}$ | 0.75[*d] |
| | | Opx-bridgmanite | 4 | 1045 | 0% | 0.30 (19) | 2.00 | $7.42\times10^{-2}$ | $8.33\times10^{-2}$ | --[*d] |
| I959R | 100 | Post-spinel[*a] | 3 | 474 | ~30% | −0.03 (11) | 0.94 | $1.18\times10^{-4}$ | $1.18\times10^{-4}$ | 1.29[*d] |
| I1105 | 5 | Post-spinel[*a] | 4 | 397 | ~30% | −0.24 (15) | 0.57 | $1.84\times10^{-4}$ | $1.84\times10^{-4}$ | 1.18[*d] |
| I1143 | 300 | Post-spinel[*a] | 3 | 456 | ~30% | 0.08 (8) | 1.21 | $1.48\times10^{-4}$ | $1.48\times10^{-4}$ | 1.85[*d] |
| I1147 | 2 | Post-spinel[*a] | 3 | 318 | ~30% | −0.33 (12) | 0.47 | $1.64\times10^{-4}$ | $1.64\times10^{-4}$ | 0.79[*d] |
| | | Opx-bridgmanite | 3 | 282 | 0% | 0.25 (15) | 1.77 | $3.99\times10^{-2}$ | $4.39\times10^{-2}$ | --[*d] |
| I1023 | 100 | Post-spinel[*a] | 3 | 447 | ~30% | 0.03 (17) | 1.08 | $2.43\times10^{-4}$ | $2.43\times10^{-4}$ | 2.16[*d] |
| | | Opx-bridgmanite | 4 | 354 | 0.0% | 0.87 (23) | 7.41 | $4.06\times10^{-2}$ | $5.66\times10^{-2}$ | --[*d] |
| | | Sol-gel-1.5 — Image #32 | 139 | | 22.3% | −0.08 (12) | 0.82 | $6.07\times10^{-5}$ | $6.07\times10^{-5}$ | 1.19 |
| | | Image #35 | 117 | | 8.7% | 0.14 (17) | 1.38 | $8.62\times10^{-4}$ | $8.76\times10^{-4}$ | 1.78 |
| | | Image #36 | 111 | | 6.8% | 0.16 (18) | 1.44 | $1.05\times10^{-3}$ | $1.09\times10^{-3}$ | 1.77 |
| | | Image #37 | 152 | | 31.7% | −0.14 (12) | 0.73 | $3.32\times10^{-5}$ | $3.32\times10^{-5}$ | 1.27 |
| | | Image #38 | 145 | | 34.5% | −0.13 (14) | 0.74 | $3.58\times10^{-5}$ | $3.58\times10^{-5}$ | 1.28 |
| | | Image #46 | 144 | | 33.5% | −0.12 (11) | 0.76 | $3.96\times10^{-5}$ | $3.96\times10^{-5}$ | 1.31 |
| | | Image #146 | 200 | | 11.4% | −0.06 (19) | 0.87 | $7.95\times10^{-5}$ | $7.94\times10^{-5}$ | 1.52 |
| | | Sol-gel-1.25 — Image #47 | 73 | | 51.8% | −0.05 (15) | 0.89 | $9.25\times10^{-5}$ | $9.25\times10^{-5}$ | --[*d] |
| | | Image #48 | 91 | | 22.2% | 0.08 (18) | 1.20 | $4.20\times10^{-4}$ | $4.20\times10^{-4}$ | 1.64 |
| | | Image #131 | 154 | | 62.9% | −0.06 (13) | 0.87 | $8.09\times10^{-5}$ | $8.09\times10^{-5}$ | --[*d] |
| | | Image #134 | 267 | | 61.8% | −0.09 (15) | 0.81 | $5.63\times10^{-5}$ | $5.63\times10^{-5}$ | --[*d] |
| | | Image #135 | 173 | | 58.2% | −0.04 (11) | 0.91 | $1.02\times10^{-4}$ | $1.02\times10^{-4}$ | --[*d] |
| | | Image #136 | 99 | | 19.9% | 0.08 (15) | 1.21 | $4.47\times10^{-4}$ | $4.47\times10^{-4}$ | 1.69 |
| | | sol-gel-1.125 — Image #138 | 53 | | 3.7% | 0.33 (18) | 2.13 | $5.89\times10^{-3}$ | $8.29\times10^{-3}$ | 2.35 |
| | | Image #54 | 140 | | 10.7% | 0.09 (19) | 1.22 | $4.65\times10^{-4}$ | $4.67\times10^{-4}$ | 1.46 |
| | | Image #57 | 68 | | ~0.0% | 0.81 (28) | 6.44 | $2.76\times10^{-2}$ | $3.76\times10^{-2}$ | --[*d] |
| I1260 | 1.5 | Post-spinel | 3 | 398 | ~30% | −0.30 (14) | 0.50 | $2.98\times10^{-4}$ | $2.98\times10^{-4}$ | 0.88[*d] |
| | | Opx-bridgmanite | 3 | 301 | 0% | 0.21 (16) | 1.62 | $4.11\times10^{-2}$ | $4.45\times10^{-2}$ | --[*d] |
| | | Glass-1.02 | 3 | 504 | 0.3% | 0.27 (19) | 1.85 | $7.13\times10^{-2}$ | $6.69\times10^{-2}$ | 3.24 |
| | | Glass-1.05 | 3 | 294 | 1.3% | 0.29 (18) | 1.95 | $1.41\times10^{-1}$ | $7.72\times10^{-2}$ | 2.63 |
| | | Glass-1.1 | 3 | 616 | 3.0% | 0.19 (19) | 1.55 | $8.03\times10^{-2}$ | $1.06\times10^{-1}$ | 2.06 |
| | | Glass-1.2 | 3 | 237 | 8.1% | −0.06 (16) | 0.87 | $5.52\times10^{-3}$ | $5.47\times10^{-3}$ | 1.11 |
| | | Glass-1.3 | 3 | 448 | 11.5% | −0.20 (16) | 0.63 | $1.01\times10^{-3}$ | $1.00\times10^{-3}$ | 0.96 |
| | | Glass-1.4 | 3 | 341 | 13.7% | −0.29 (13) | 0.51 | $3.44\times10^{-4}$ | $3.43\times10^{-4}$ | 0.83 |
| | | Glass-1.5 | 4 | 300 | 18.6% | −0.18 (17) | 0.67 | $1.37\times10^{-3}$ | $1.37\times10^{-3}$ | 1.02 |
| I1266 | 10 | Post-spinel | 3 | 360 | ~30% | −0.21 (16) | 0.62 | $1.38\times10^{-4}$ | $1.38\times10^{-4}$ | 1.19[*d] |
| | | Opx-bridgmanite | 4 | 213 | 0% | 0.42 (15) | 2.66 | $2.44\times10^{-2}$ | $2.87\times10^{-2}$ | --[*d] |
| | | Glass-1.02 | 4 | 252 | 0.4% | 0.42 (19) | 2.66 | $3.73\times10^{-2}$ | $2.86\times10^{-2}$ | 4.14 |
| | | Glass-1.05 | 3 | 231 | 1.2% | 0.27 (15) | 1.88 | $1.81\times10^{-2}$ | $1.05\times10^{-2}$ | 2.24 |
| | | Glass-1.1 | 3 | 317 | 3.1% | 0.18 (17) | 1.53 | $1.16\times10^{-2}$ | $1.50\times10^{-2}$ | 1.60 |
| | | Glass-1.2 | 3 | 450 | 8.8% | −0.01 (16) | 0.99 | $1.56\times10^{-3}$ | $1.56\times10^{-3}$ | 1.25 |
| | | Glass-1.3 | 3 | 345 | 10.7% | −0.06 (18) | 0.87 | $8.00\times10^{-4}$ | $7.98\times10^{-4}$ | 1.35 |
| | | Glass-1.4 | 3 | 549 | 15.0% | −0.06 (22) | 0.88 | $8.46\times10^{-4}$ | $8.45\times10^{-4}$ | 1.44 |
| | | Glass-1.5 | 3 | 470 | 18.3% | −0.20 (14) | 0.63 | $1.57\times10^{-4}$ | $1.57\times10^{-4}$ | 1.05 |
| I1269 | 100 | Post-spinel | 4 | 244 | ~30% | −0.00 (14) | 0.99 | $1.62\times10^{-4}$ | $1.62\times10^{-4}$ | 1.50[*d] |
| | | Opx-bridgmanite | 6 | 400 | 0% | 0.86 (19) | 7.18 | $3.72\times10^{-2}$ | $5.15\times10^{-2}$ | --[*d] |
| | | Glass-1.02 | 3 | 302 | 0.4% | 0.81 (17) | 6.42 | $5.76\times10^{-2}$ | $3.71\times10^{-2}$ | 7.97 |
| | | Glass-1.05 | 3 | 267 | 1.1% | 0.72 (19) | 5.30 | $7.93\times10^{-2}$ | $2.13\times10^{-2}$ | 4.73 |
| | | Glass-1.1 | 3 | 233 | 3.5% | 0.47 (18) | 2.98 | $2.76\times10^{-2}$ | $4.73\times10^{-2}$ | 2.61 |
| | | Glass-1.2 | 3 | 525 | 9.0% | 0.23 (19) | 1.69 | $2.43\times10^{-3}$ | $2.48\times10^{-3}$ | 2.17 |
| | | Glass-1.3 | 3 | 322 | 10.4% | 0.09 (15) | 1.23 | $4.91\times10^{-4}$ | $4.93\times10^{-4}$ | 1.43 |
| | | Glass-1.4 | 3 | 224 | 14.6% | 0.13 (17) | 1.35 | $7.80\times10^{-4}$ | $7.81\times10^{-4}$ | 1.79 |
| | | Glass-1.5 | 3 | 520 | 18.5% | 0.11 (19) | 1.30 | $6.38\times10^{-4}$ | $6.38\times10^{-4}$ | 2.06 |

[*a]: Already reported in our recent paper[37].

[*b]: Based on continuous change of $n$ with $X_{fpc}$.

[*c]: Based on discontinuous change of $n$ with $X_{fpc}$.

[*d]: $\bar{r}$ is invalid because of either $X_{fpc}$ = 0% or significant/complete interconnections of ferropericlase grains.

The $d$ was obtained from two-dimensional BSE-images without 3D correction. All experiments were performed at 27 GPa and 2200 K.

**Extended Data Table 3 | Composition of bridgmanite analyzed by EPMA**

| Sample | N | MgO (wt.%) | SiO$_2$ (wt.%) | FeO (wt.%) | Total (wt.%) | Mg (atomic) | Si (atomic) | Fe (atomic) | Fe/(Mg+Fe) ratio (%) |
|---|---|---|---|---|---|---|---|---|---|
| I1269_opx-bridgmanite | 11 | 35.46 (36) | 58.18 (45) | 6.90 (14) | 100.55 (71) | 0.906 (6) | 0.997 (3) | 0.099 (2) | 9.85 (21) |
| I1269_glass-1.02 | 13 | 35.95 (60) | 58.54 (58) | 6.63 (25) | 101.12 (89) | 0.912 (12) | 0.997 (6) | 0.094 (4) | 9.37 (33) |
| I1269_glass-1.05 | 12 | 36.91 (40) | 57.87 (88) | 6.63 (37) | 101.41 (87) | 0.936 (13) | 0.985 (7) | 0.094 (5) | 9.15 (48) |
| I1269_glass-1.1 | 11 | 36.51 (49) | 58.46 (43) | 6.07 (19) | 101.04 (58) | 0.926 (9) | 0.994 (5) | 0.086 (3) | 8.54 (29) |
| I1269_glass-1.2 | 5 | 37.86 (32) | 57.97 (52) | 5.05 (33) | 100.88 (24) | 0.959 (10) | 0.985 (5) | 0.072 (5) | 6.95 (45) |
| I1269_glass-1.3 | 43 | 37.69 (91) | 58.21 (73) | 5.35 (45) | 101.25 (102) | 0.952 (18) | 0.986 (9) | 0.076 (7) | 7.37 (63) |
| I1269_glass-1.4 | 23 | 38.25 (82) | 58.10 (83) | 4.51 (54) | 100.85 (60) | 0.966 (21) | 0.985 (11) | 0.064 (8) | 6.20 (74) |
| I1269_glass-1.5 | 26 | 38.69 (72) | 58.30 (91) | 4.63 (29) | 101.62 (101) | 0.971 (16) | 0.982 (9) | 0.065 (4) | 6.29 (37) |
| I1266_opx-bridgmanite | 12 | 35.68 (47) | 58.19 (45) | 7.03 (54) | 100.90 (57) | 0.909 (8) | 0.995 (5) | 0.101 (8) | 9.95 (72) |
| I1266_glass-1.02 | 12 | 36.28 (44) | 58.08 (72) | 6.76 (55) | 101.12 (57) | 0.923 (10) | 0.991 (8) | 0.096 (8) | 9.46 (70) |
| I1266_glass-1.05 | 12 | 36.95 (24) | 57.88 (45) | 6.57 (25) | 101.40 (53) | 0.937 (7) | 0.985 (3) | 0.094 (4) | 9.07 (35) |
| I1266_glass-1.1 | 7 | 36.32 (97) | 57.97 (104) | 6.76 (71) | 101.05 (94) | 0.924 (23) | 0.990 (11) | 0.097 (11) | 9.46 (105) |
| I1260_opx-bridgmanite | 13 | 36.45 (49) | 57.94 (65) | 6.66 (72) | 101.06 (42) | 0.927 (12) | 0.989 (7) | 0.095 (11) | 9.30 (97) |
| I1260_glass-1.02 | 13 | 35.23 (40) | 58.33 (37) | 7.01 (19) | 100.56 (59) | 0.900 (8) | 1.000 (4) | 0.100 (3) | 10.04 (28) |
| I1260_glass-1.05 | 16 | 36.68 (57) | 57.98 (67) | 6.49 (65) | 101.15 (78) | 0.932 (13) | 0.988 (6) | 0.093 (9) | 9.03 (88) |
| I1260_glass-1.1 | 13 | 36.75 (53) | 57.74 (90) | 6.34 (60) | 100.82 (72) | 0.936 (15) | 0.987 (9) | 0.091 (9) | 8.82 (78) |
| I930_opx-bridgmanite | 10 | 35.56 (61) | 58.22 (89) | 7.43 (21) | 101.20 (97) | 0.905 (14) | 0.994 (7) | 0.106 (4) | 10.49 (34) |
| I930-post-spinel | 8 | 38.39 (57) | 58.42 (62) | 4.29 (47) | 101.10 (42) | 0.966 (14) | 0.987 (8) | 0.061 (7) | 5.89 (63) |
| I928_opx-bridgmanite | 13 | 35.89 (65) | 58.38 (50) | 7.06 (24) | 101.32 (88) | 0.911 (12) | 0.994 (6) | 0.101 (4) | 9.94 (36) |
| I951_opx-bridgmanite | 11 | 34.64 (69) | 57.55 (82) | 7.07 (51) | 99.26 (60) | 0.897 (19) | 1.000 (11) | 0.103 (7) | 10.26 (62) |
| I1147_opx-bridgmanite | 13 | 34.25 (65) | 58.38 (29) | 7.05 (57) | 99.68 (32) | 0.882 (13) | 1.008 (3) | 0.102 (9) | 10.37 (92) |
| I1143_ post-spinel | 10 | 38.04 (70) | 58.13 (90) | 4.16 (34) | 100.33 (95) | 0.964 (14) | 0.988 (8) | 0.059 (5) | 5.78 (49) |
| I1023_opx-bridgmanite | 19 | 35.10 (73) | 57.54 (86) | 7.13 (24) | 99.77 (121) | 0.905 (15) | 0.996 (7) | 0.103 (4) | 10.24 (40) |
| I1023_post-spinel | 4 | 38.22 (70) | 58.13 (62) | 4.13 (18) | 100.48 (89) | 0.967 (13) | 0.987 (7) | 0.059 (3) | 5.72 (24) |
| I1023_sol-gel-1.125 | 17 | 37.09 (92) | 57.96 (67) | 4.87 (90) | 99.92 (72) | 0.946 (17) | 0.992 (6) | 0.070 (14) | 6.88 (133) |
| I1023_sol-gel-1.25 | 14 | 36.18 (86) | 57.37 (78) | 6.43 (75) | 99.97 (76) | 0.930 (19) | 0.989 (7) | 0.093 (11) | 9.07 (112) |
| I1023_sol-gel-1.5 | 13 | 37.35 (96) | 57.18 (62) | 5.13 (68) | 99.65 (90) | 0.958 (17) | 0.984 (7) | 0.074 (10) | 7.16 (99) |

Note that not all the samples in Extended Data Table 2 are analyzed because the analysis of bridgmanite is significantly affected by the neighboring ferropericlase grains when the grain size is small and $X_{fpc}$ is high. The data in parentheses are one standard deviation of the N analyzed points. The atomic number is normalized to O=3.

## Extended Data Table 4 | The parameters for calculation of $D^{lat}$ and $\delta D^{gb}$ based on ref. 58

|  | $D^{lat}$ | $\delta D^{gb}$ |
|---|---|---|
| $D_0^{lat}$, $\delta D_0^{gb}$ | 2.74×10⁻¹⁰ m²/s (ref.[58]) | 7.12×10⁻¹⁷ m³/s (ref.[58]) |
| $\Delta H^{lat}$, $\Delta H^{gb}$ at 25 GPa | 336 (37) kJ/mol (ref.[58]) | 311 (48) kJ/mol (ref.[58]) |
| $\Delta V^{lat}$, $\Delta V^{gb}$ | 1.7 (0.4) cm³/mol (ref.[61]) | 4.0 (0.7) cm³/mol (ref.[62]) |
| At 27 GPa, 2200 K | 2.39×10⁻¹⁸ m²/s | 1.90×10⁻²⁴ m³/s |

The $D^{lat}$ and $\delta D^{gb}$ were measured in ref. 58. at a pressure of 25 GPa and temperatures of 1673 – 2073 K. They were corrected to various pressure and temperature conditions using the Arrhenius equations: $D^{lat} = D_0 \exp(-(\triangle H^{lat} + (P - 25)\triangle V^{lat})/RT)$, $\delta D^{gb} = \delta D_0 \exp(-(\triangle H^{gb} + (P - 25)\triangle V^{gb})/RT)$, where $\Delta H^{lat}$ and $\Delta H^{gb}$ are the activation enthalpies at 25 GPa, $\Delta V^{lat}$ and $\Delta V^{gb}$ are the activation volumes, and $D_0^{lat}$, $\delta D_0^{gb}$ are pre-exponential factors for $D^{lat}$ and $\delta D^{gb}$, respectively, $P$ is the pressure, $R$ is the ideal gas constant, and $T$ is the temperature.