## [Peer Review File · Nature]

Manuscript Title: Bridgmanite grain size variation accounts for the mid-mantle viscosity jump

Editorial Notes:

Redactions – Third Party Material

Reviewer Comments & Author Rebuttals

Reviewer Reports on the Initial Version:

Referee #1:

Review comments on "Bridgmanite grain size variation accounts for the mid-mantle viscosity jump" by Hongzhan Fei et al. submitted to Nature

This manuscript reports a study of effect on ferropericlasite content on grain-growth in bridgmanite-ferropericlasite aggregate based on high-pressure and high-temperature experiments. Based on the experimental results, the authors discuss variation of grain-size and viscosity in the Earth's lower mantle. The discussion is original and of broad interests. However, there are some crucial questions on the experimental results. In this study, results of the solution-gelation-derived (sol-gel) bridgmanite-ferropericlasite aggregates are key for determination of grain-growth law as a function of ferropericlasite volume fraction. There is significant heterogeneity in ferropericlasite fraction in these samples and its origin is unclear. Only one experimental run is reported for the sol-gel aggregates. More runs are desirable to test reproducibility. Detailed comments are as follows.

1.

Ferropericlasite volume fraction (fpc) in the solution-gelation-derived bridgmanite-ferropericlasite mixtures varied significantly even in the single specimen (Ext. Data Table 2): fpc = 6.8-34.5% for sol-gel-1.5, 22.2-62.9% for sol-gel-1.25, and 0.0-10.7% for sol-gel-1.125. However, reason of this large variation of fpc is not explained at all in the manuscript. It is essentially important to investigate why and when does this fpc heterogeneity occur.

It is not clear whether the fpc heterogeneity exists or not before annealing and after synthesis. If the fpc heterogeneity developed during grain-growth annealing by segregation of ferropericlasite, interpretation of grain-growth kinetics would be more complicated or impossible. Because the analysis of the fpc dependent grain-growth law highly depends on results of the sol-gel derived samples (see Fig. 3), accuracy of these results is critically important for the later discussion.

2.

Accurate information of Fe content in the samples is lacking in the manuscript including supplementary materials. Description of Fe content in the starting materials and the annealed samples are necessary. Yamazaki et al. (2009) and Fei et al. (2021) showed that presence of Fe enhances grain-growth in bridgmanite-(ferro)periclasite aggregate. I am also curious about homogeneity of Fe content in sol-gel derived samples because fpc is highly heterogeneous in these samples.

Yamazaki et al. (2009) Phys. Earth Planet. Int. 174, 138-144.

Minor comments

Eqs. (2) and (3): These equations and parameters (A, B, C, A', B', and C') hold when unit of grain size d is μm and unit of k is $\mu\text{m}^n/\text{s}$, respectively. The units of d and k for these equations should be described explicitly.

Lines 97-98: The authors say "the grain growth of bridgmanite in the single-phase system is three orders of magnitude faster" based on comparison of k values for ferropericlase content (fpc) = 0% and 10-60%. However, this is not accurate because unit of k value is different for fpc = 0% and 10-60%. The unit of k value is $\mu\text{m}^n/\text{s}$ and n value changes depending on fpc. It is meaningless to compare these k values themselves because they are in different unit.

Line 326: "0.15 μm " may be "0.15 mm".

Line 355: "found i the post-spinel" may be "found in the post-spinel".

Referee #2 (Joerg Renner):

[Please see attached PDF files below]

Dear editor,

You asked me to review a manuscript entitled “Bridgmanite grain size variation accounts for the mid-mantle viscosity jump” authored by Hongzhan Fei, Maxim D. Ballmer, Ulrich Faul, Nicolas Walte, and Tomoo Katsura and submitted to nature communications. The authors use their experimental results for the grain-growth kinetics in synthetic, polycrystalline bridgmanite aggregates with different amounts of ferropericlas to discuss the potential origin of an increase in viscosity with increasing depth in the lower mantle, indicated by, .e.g., some geophysical observations. Scientists in only few laboratories worldwide are capable of performing the type of experiments, for which the results are presented. The experiments are clearly important for our understanding of the microstructural state of the lower mantle and its consequences for dynamic processes and I am more than happy to praise the authors for their experimental efforts and accomplishments. However, in its present form, I do not consider the manuscript suitable for publication owing to a lack of conciseness and precision in reasoning as well as –that may well be my personal bias- deficiencies in structuring and presentation.

Grain size is undoubtedly a critical internal variable for the viscosity of polycrystalline aggregates. Yet, boldly speaking, the authors have to leave it to chance rather than some geodynamical evolution that the current distribution of grain size (actually dictated by a fortuitous ferropericlas-depth distribution in their limited model realm) results in a viscosity jump at the right depth. To me, the manuscript suffers from an unbalanced consideration of alternatives, to the extent that they are not even mentioned. The followed string of logic is totally devoid of an uncertainty analyses.

Please, find detailed comments below and attached a digitally annotated pdf-version of the manuscript. My comments in the pdf-file reflect my immediate responses when reading the manuscript and did not experience any “post”-polishing for “diplomacy” or the like; they may thus appear frank in cases but no offense meant whatsoever. Thank you very much for providing me the chance to read about this study and to think about lower-mantle dynamics. Please, do not hesitate to contact me in case of questions or problems.

Kind regards,
Joerg

study set-up/objective:

The viscosity of polycrystalline aggregates depends on thermodynamic state variables, i.e., pressure, temperature, stress, activities (affected by composition), ... , and as well as an (micro-)structural parameters, e.g., grain size (affected by composition), and through either of them on the activated deformation mechanism. Thus, explaining a viscosity variation in one of Earth’s shells requires considering variations in all these variables as potential causes. The authors focus on grain size because micromechanical models for diffusion creep suggest a power-law dependence of viscosity on grain size.

approach:

The authors constrain grain-growth kinetics in polycrystalline aggregates composed of bridgmanite and ferropericlas, here the second phase, by annealing experiments (performed at conditions of the lower mantle) and subsequent microstructural analyses. They interpret their data using a “generic” growth law of the form:

$$d^n - d_0^n = kt \Rightarrow d^n \simeq kt \text{ for } d \gg d_0,$$

where the simplifying step of „neglecting starting grain size” allows them to evaluate the model parameters, growth exponent n and kinetics constant k , from linear relations between logarithmic values:

$$\log d = \frac{1}{n}(\log k + \log t).$$

To me, the expectations regarding the growth exponent for different rate-controlling processes are insufficiently discussed. An in-depth discussion of the empirically constrained values of ~ 3 for “pure” aggregates and ~ 5 for aggregates containing ferropericlas is missing. From my point of view, this lack is problematic in two ways:

1) To the best of my knowledge, a model comprising growth exponents continuously changing with second-phase content has not been presented so far. In the contrast, micromechanical models give growth exponents that change discontinuously depending on the role of the secondary phases. In this regard, I do not see the value of the performed empirical fitting:

$$n = A' \exp\left(\frac{\text{fpc}\%}{B'}\right) + C'$$

$$k = A'' \exp\left(\frac{\text{fpc}\%}{B''}\right) + C''$$

In addition, the authors require

$$\log d = A \exp\left(\frac{\text{fpc}\%}{B}\right) + C$$

at the same time. While numerical fitting allows them to get the three sets of fit parameters independently, from an analytical perspective only two should be independent. I cannot see how you can request these three equations to hold at the same time; once n and k are constrained, $(\log) d$ follows from the underlying kinetics law:

$$d^n \approx kt \Rightarrow$$

$$d^{A' \exp\left(\frac{\text{fpc}\%}{B'}\right) + C'} = \left[A'' \exp\left(\frac{\text{fpc}\%}{B''}\right) + C'' \right] t_{\text{fixed}}$$

I cannot see a simple way to confirm the assertion that $\log d$ follows a similar form as n and k :

$$\Rightarrow \left[A' \exp\left(\frac{\text{fpc}\%}{B'}\right) + C' \right] \log d = \log \left[A \exp\left(\frac{\text{fpc}\%}{B}\right) + C \right]$$

$$\Leftrightarrow \log d = \frac{1}{A' \exp\left(\frac{\text{fpc}\%}{B'}\right) + C'} \log \left[A \exp\left(\frac{\text{fpc}\%}{B}\right) + C \right].$$

This expression should be recast to

$$\log d = A \exp\left(\frac{\text{fpc}\%}{B}\right) + C,$$

which is not at all obvious.

The “strange” fitting exercise seems obsolete to me, given that the authors never use their results in the subsequent reasoning about lower-mantle viscosity (if I am not mistaken).

2) The authors’ disregard of analytical/micromechanical models for grain growth appears in stark contrast to their “uncritical belief” in creep laws (eqs. 3 and 4 in appendix). The manuscript lacks a

critical discussion of either the model equations or the used parameters (and their uncertainties). This lack becomes critically important at the point of the “logical thread”, where the authors realize that the predictions for a potential viscosity jump actually critically depend on the (quantitative predictions regarding) dislocation creep behavior. The role of pressure on diffusion kinetics should, for example, be addressed. What kind of activation volume would be required to observe a change from “thermal activation” to “pressure deactivation”.

reasoning

As intriguing as grain size may be as the cause for the viscosity jump, it brings a few problems with it. Grain size is an “unbounded” internal variable (as long as dynamic recrystallization, irreversible thermodynamics, is excluded, see below); the static thermodynamic equilibrium of the mantle corresponds to single crystals of the involved phases. Thus, one has to answer the central question why the current micro-structural state is such that the viscosity jump results. In this regard, the manuscript remains shallow. I cannot identify convincing arguments why the ferropericlas concentration should vary with depth in the required form (it is this content that in the authors’ reasoning controls grain size).

The authors neglect that the particles’ content is only one part of the role of secondary phases on grain growth. Actually, interparticle spacing controls the spatial length for unhindered growth. Quantification of interparticle spacing requires at list the incorporation of particle size in addition to content. Thus, interparticle spacing should be determined and reported. The authors’ conceptual focus on “particle content” leads to the critical omission of Ostwald ripening of secondary particles. The interparticle spacing of ferropericlas in the mantle, the critical length scale for grain size of bridgmanite, is not discussed whatsoever. Four billion years is a long time even for a process that one might consider “slow” –on whatever scale.

As the authors realize, whether grain size can account for a viscosity jump critically depends on the potential to activate deformation mechanisms alternative to diffusion creep. Dislocation creep limits the grain-size effect. Grain-growth kinetics alone do not allow one to make predictions on viscosity variations –as apparently “implied” by the study- but all reasoning regarding viscosity critically hinges on the dislocation-creep law, outside of the “research” of the authors. To some extent, the false impression is created that studying grain growth provides the rheology of the mantle. Quantitative estimates of viscosity for the mantle hinge on constraints on flaw laws rather than on anything constrained by this study alone. In addition to these conceptual issues, the relevant processes may change when dislocation creep is active activation. When operating for sufficiently large strains, dislocation creep is associated with “dynamic recrystallization”. Then, the grain size is not controlled by the growth law alone but grain size may assume a “steady-state” value dictated by the power exerted on the mantle volume.

writing:

Please see the digital annotations indicating where I think the wording lacks conciseness.

formal aspects:

To me, the structure of the manuscript is somewhat odd. The authors seem to spend more “time”, i.e., more text, on discarding alternative reasons for changes in viscosity than for the grain-size related ones advocated by them. Actually, the discard of previously presented alternative, to me, requires a concise presentation in the introduction, rather than in the discussion, as part of the “research-gap identification” motivating the study of grain-growth kinetics.

Author Rebuttals to Initial Comments:

A list of responses to reviewers' comments

We sincerely appreciate the anonymous reviewer and Prof. Jörg Renner, whose comments are very helpful for us to improve our manuscript.

Following the reviewers' comments, we have significantly revised the manuscript. In case if we misunderstand some comments, we would be happy to improve the manuscript by the reviewers' further suggestions.

The main concern of reviewer #1 is that the heterogeneity of sol-gel samples may have affected the experimental results. Therefore, we performed additional experiments using silicate glasses as starting materials, which produced homogenously distributed bridgmanite + ferropericlase samples. As shown below, the glass and sol-gel samples show consistent results (Fig. 3).

The key point from reviewer #2 is that the uncertainty should be discussed when simulating the viscosity. The uncertainty analyses are now provided (Line 415-442). The major uncertainties come from diffusion coefficients. However, these uncertainties only affect the absolute values of the simulated viscosity, but do not affect the viscosity contrast between bridgmanite-enriched and pyrolitic rocks. This is because the viscosity contrast is controlled by the ratio of creep rate in bridgmanite-enriched rocks and pyrolite rocks, and this ratio is independent from diffusion coefficients (Line 415-427).

The second point from reviewer #2 is that our data fitting process may be inappropriate because the grain size exponent (n) may change with X_{fpc} discontinuously. Because the $n - X_{fpc}$ relation is unknown, we refitted the $n - X_{fpc}$ relation based on both discontinuous and continuous n from 2.9 to 5.2 with increasing X_{fpc} from 0 to 30% and recalculated the grain-growth rate using these two $n - X_{fpc}$ relations (Line 99-104; Fig. 3, 4; Line 305-311).

Reviewer #2 also pointed out the importance of dislocation creep on the viscosity contrast, and that dislocation creep rate is not related to grain size. The related discussion is given in Line 121-129; Line 415-432. The grain size contrast causes the diffusion creep rate of pyrolitic rocks to be more than four orders of magnitude higher than the bridgmanite-enriched rocks. By adding the grain-size independent dislocation creep rate, the total creep rate of pyrolitic rocks remains 1 – 2 orders of magnitude higher (Fig. 4B). Therefore, the viscosity contrast is still mainly caused by the grain-size contrast (Line 121-129). This is because the ratio of creep rates between bridgmanite-enriched rocks and pyrolitic rocks is strongly controlled by grain size even if dislocation creep dominates in the bridgmanite-enriched rocks (Line 415-432).

The following is a detail list of our response to reviewers' comments. The manuscript is revised accordingly.

=====

Referee #1:

Review comments on “Bridgmanite grain size variation accounts for the mid-mantle viscosity jump” by Hongzhan Fei et al. submitted to Nature

This manuscript reports a study of effect on ferropericlasite content on grain-growth in bridgmanite-ferropericlasite aggregate based on high-pressure and high-temperature experiments. Based on the experimental results, the authors discuss variation of grain-size and viscosity in the Earth’s lower mantle. The discussion is original and of broad interests. However, there are some crucial questions on the experimental results. In this study, results of the solution-gelation-derived (sol-gel) bridgmanite-ferropericlasite aggregates are key for determination of grain-growth law as a function of ferropericlasite volume fraction. There is significant heterogeneity in ferropericlasite fraction in these samples and its origin is unclear. Only one experimental run is reported for the sol-gel aggregates. More runs are desirable to test reproducibility. Detailed comments are as follows.

It’s true that there is significant heterogeneity in the sol-gel samples. We performed additional runs using uniform silicate glass samples with various bulk compositions, i.e., $(\text{Mg,Fe})_x\text{SiO}_{2+x}$ with $x = 1.02, 1.05, 1.1, 1.2, 1.3,$ and 1.5 . As shown below and in Ext. Data Fig. 1D, the samples from glass showed uniform distributions of ferropericlasite. The grain sizes of the run products from glass, sol-gel, olivine, and opx agree well with each other (Line 74-75; Fig. 3A-C).

The heterogeneity in the sol-gel samples occurred before grain-growth experiments, probably because of inhomogeneous Si distribution during gelation (Line 361-362). As shown in Ext. Data Fig. 1C, the sol-gel samples after pre-synthesis already show domains with different X_{fpc} contents, but within each domain the X_{fpc} is rather homogenous. These observations suggest that results from different domains can represent the grain growth at various X_{fpc} because atomic diffusion in bridgmanite is extremely slow and therefore the X_{fpc} in each domain should have remained unchanged during the grain growth experiment (Line 383-387).

Sol-gel sample after pre-synthesis but before grain-growth experiment (Extended Data Fig. 3C)

1. Ferropericlasite volume fraction (fpc) in the solution-gelation-derived bridgmanite-ferropericlasite mixtures varied significantly even in the single specimen (Ext. Data Table 2): fpc = 6.8-34.5% for sol-gel-1.5, 22.2-62.9% for sol-gel-1.25, and 0.0-10.7% for sol-gel-1.125. However, reason of this large variation of fpc is not explained at all in the manuscript. It is essentially important to investigate why and when does this fpc heterogeneity

occur. It is not clear whether the fpc heterogeneity exists or not before annealing and after synthesis. If the fpc heterogeneity developed during grain-growth annealing by segregation of ferropericlase, interpretation of grain-growth kinetics would be more complicated or impossible. Because the analysis of the fpc dependent grain-growth law highly depends on results of the sol-gel derived samples (see Fig. 3), accuracy of these results is critically important for the later discussion.

As explained above, the heterogeneity already occurred in the synthesis procedure before grain-growth experiment (Ext. Data Fig. 1C; Line 361-362). After the pre-synthesis, the grains already reached the equilibrated texture (120degree triple junction). Additionally, the glass and sol-gel samples show identical results. Therefore, the heterogeneity in the sol-gel samples should not affect the grain-growth experiments (Fig. 3A; Line 383-387).

2. Accurate information of Fe content in the samples is lacking in the manuscript including supplementary materials. Description of Fe content in the starting materials and the annealed samples are necessary. Yamazaki et al. (2009) and Fei et al. (2021) showed that presence of Fe enhances grain-growth in bridgmanite-(ferro)periclase aggregate. I am also curious about homogeneity of Fe content in sol-gel derived samples because fpc is highly heterogeneous in these samples.

Yamazaki et al. (2009) Phys. Earth Planet. Int. 174, 138-144.

The Fe contents were measured and are listed in Ext. Data Table 3. Only samples with relatively large grain sizes are analyzed due to the limited spatial resolution of EPMA. We did not find clear heterogeneity of Fe% in the samples. The Mg/Fe in the starting material was about 9/1.

Fei et al. (2021) and Yamazaki et al. (2009) show that the difference of $\log d$ between Fo90 and Fo100 samples is only 0.1 log unit. In contrast, the $X_{fpc} = 0\%$ samples have 0.7-1.0 log unit larger grain sizes than $X_{fpc} = 30\%$ samples. Additionally, our sol-gel samples did not show clear heterogeneity of Fe%. Therefore, the decrease of $\log d$ with X_{fpc} should not be caused by a variation in the Fe content (Line 95-98).

[REDACTED]

Minor comments

Eqs. (2) and (3): These equations and parameters (A, B, C, A', B', and C') hold when unit of grain size d is μm and unit of k is $\mu\text{m}^n/\text{s}$, respectively. The units of d and k for these equations should be described explicitly.

The units are described ($\mu\text{m}^n/\text{s}$) (Line 103; Fig. 3E).

Lines 97-98: The authors say "the grain growth of bridgmanite in the single-phase system is three orders of magnitude faster" based on comparison of k values for ferropericlase content (fpc) = 0% and 10-60%. However, this is not accurate because unit of k value is different for fpc = 0% and 10-60%. The unit of k value is $\mu\text{m}^n/\text{s}$ and n value changes depending on fpc. It is meaningless to compare these k values themselves because they are in different unit.

The related sentence is deleted in the revised manuscript.

Line 326: "0.15 μm " may be "0.15 mm".

"0.15 μm " is corrected to "0.15 mm" (Line 350-351).

Line 355: "found i the post-spinel" may be "found in the post-spinel".

"i" is corrected to "in" (Line 379).

=====

Referee #2 (Joerg Renner):

You asked me to review a manuscript entitled “Bridgmanite grain size variation accounts for the mid-mantle viscosity jump” authored by Hongzhan Fei, Maxim D. Ballmer, Ulrich Faul, Nicolas Walte, and Tomoo Katsura and submitted to nature communications. The authors use their experimental results for the grain-growth kinetics in synthetic, polycrystalline bridgmanite aggregates with different amounts of ferropericlas to discuss the potential origin of an increase in viscosity with increasing depth in the lower mantle, indicated by, .e.g., some geophysical observations. Scientists in only few laboratories worldwide are capable of performing the type of experiments, for which the results are presented. The experiments are clearly important for our understanding of the microstructural state of the lower mantle and its consequences for dynamic processes and I am more than happy to praise the authors for their experimental efforts and accomplishments. However, in its present form, I do not consider the manuscript suitable for publication owing to a lack of conciseness and precision in reasoning as well as –that may well be my personal bias- deficiencies in structuring and presentation.

Grain size is undoubtedly a critical internal variable for the viscosity of polycrystalline aggregates. Yet, boldly speaking, the authors have to leave it to chance rather than some geodynamical evolution that the current distribution of grain size (actually dictated by a fortuitous ferropericlas-depth distribution in their limited model realm) results in a viscosity jump at the right depth. To me, the manuscript suffers from an unbalanced consideration of alternatives, to the extent that they are not even mentioned. The followed string of logic is totally devoid of an uncertainty analyses.

As explained below, the ferropericlas-depth distribution in the lower mantle are explained in more detail in Line 38-43. The discussions of alternatives are simplified for balance and moved to the introduction section (Line 46-53). The uncertainty analyses of the simulations are given (Line 415-442).

With additional experiments, all the data are refitted for better precision (Line 99-104; Fig. 3). The text sections that lack conciseness and precision according to reviewer #2 are revised. The structure and presentation of the manuscript are reorganized.

Please, find detailed comments below and attached a digitally annotated pdf-version of the manuscript. My comments in the pdf-file reflect my immediate responses when reading the manuscript and did not experience any “post”-polishing for “diplomacy” or the like; they may thus appear frank in cases but no offense meant whatsoever. Thank you very much for providing me the chance to read about this study and to think about lower-mantle dynamics. Please, do not hesitate to contact me in case of questions or problems.

All the suggestions and corrections in the digital files are accordingly revised. Because they are relatively minor corrections about wording, we did not list them here, but we keep all the tracked changes in the revised manuscript.

study set-up/objective:

The viscosity of polycrystalline aggregates depends on thermodynamic state variables, i.e., pressure, temperature, stress, activities (affected by composition), ... , and as well as an (micro-)structural parameters, e.g., grain size (affected by composition), and through either of them on the activated deformation mechanism. Thus, explaining a viscosity variation in one of Earth’s shells requires considering variations in all these variables as potential causes. The authors focus on grain size because micromechanical models for diffusion creep suggest a power-law dependence of viscosity on grain size.

approach:

The authors constrain grain-growth kinetics in polycrystalline aggregates composed of bridgmanite and ferropericlas, here the second phase, by annealing experiments (performed at conditions of the lower mantle) and subsequent microstructural analyses. They interpret their data using a “generic” growth law of the form:

$$d^n - d_0^n = kt \Rightarrow d^n \approx kt \text{ for } d \gg d_0,$$

where the simplifying step of „neglecting starting grain size” allows them to evaluate the model parameters, growth exponent n and kinetics constant k , from linear relations between logarithmic values:

$$\log d = \frac{1}{n}(\log k + \log t).$$

To me, the expectations regarding the growth exponent for different rate-controlling processes are insufficiently discussed. An in-depth discussion of the empirically constrained values of ~ 3 for “pure” aggregates and ~ 5 for aggregates containing ferropericlas is missing.

A discussion about $n=3$ and $n=5$ is added (Line 84-89).

“These two values of n agree well with those inferred from theoretical models, i.e., $n = 2 \sim 3$ for grain-boundary-diffusion controlled grain-growth in a single-phase system and $n = 4 \sim 5$ for a two-phase system^{31,32}, and are comparable with those reported for other mantle minerals such as olivine, wadsleyite, and ringwoodite (single phase)³³⁻³⁵, as well as olivine-pyroxene and forsterite-nickel aggregates (two phases)^{36,37}.”

From my point of view, this lack is problematic in two ways:

1) To the best of my knowledge, a model comprising growth exponents continuously changing with second-phase content has not been presented so far. In the contrast, micromechanical models give growth exponents that change discontinuously depending on the role of the secondary phases. In this regard, I do not see the value of the performed empirical fitting:

$$n = A' \exp\left(\frac{fpc\%}{B'}\right) + C'$$

$$k = A'' \exp\left(\frac{fpc\%}{B''}\right) + C''$$

By following the suggestions, we performed new fittings with n increase discontinuously with increasing X_{fpc} . Since the $n - X_{fpc}$ relation is unknown, we refitted the data based on both continuous and discontinuous n (Fig. 3D).

Because the experimental results (Fig. 3A-C) show that d decreases continuously with increasing X_{fpc} , k is expected to decrease continuously, at least nearly continuously. Therefore, we first fit n by both continuous and discontinuous models (Fig. 3D), then calculate k from $d^n=kt$ based on both continuous and discontinuous n (Ext. Data Table 2). Finally, we fit the two sets of k to the continuous equation (Fig. 3E). It is found that the fittings based on discontinuous n and continuous n have negligible differences (solid and dashed lines in Fig. 3A-C; Fig. 3E).

In addition, the authors require

$$\log d = A \exp\left(\frac{fpc\%}{B}\right) + C$$

at the same time. While numerical fitting allows them to get the three sets of fit parameters independently, from an analytical perspective only two should be independent. I cannot see how you can request these

three equations to hold at the same time; once n and k are constrained, $(\log) d$ follows from the underlying kinetics law:

$$d^n = kt \Rightarrow$$

$$d^{A' \exp\left(\frac{fpc\%}{B'}\right) + C'} = \left[A^n \exp\left(\frac{fpc\%}{B^n}\right) + C^n \right] t_{\text{fixed}}$$

I cannot see a simple way to confirm the assertion that $\log d$ follows a similar form as n and k :

$$\Rightarrow \left[A' \exp\left(\frac{fpc\%}{B'}\right) + C' \right] \log d = \log \left[A \exp\left(\frac{fpc\%}{B}\right) + C \right]$$

$$\Leftrightarrow \log d = \frac{1}{A' \exp\left(\frac{fpc\%}{B'}\right) + C'} \log \left[A \exp\left(\frac{fpc\%}{B}\right) + C \right]$$

This expression should be recast to

$$\log d = A \exp\left(\frac{fpc\%}{B}\right) + C$$

which is not at all obvious.

The “strange” fitting exercise seems obsolete to me, given that the authors never use their results in the subsequent reasoning about lower-mantle viscosity (if I am not mistaken).

We agree that the reviewer’s model is mathematically more accurate. By following the reviewer’s suggestion, we refitted the data (Fig. 3). We initially fit $\log d$ to the empirical equation because $\log d$ is directly from experiments so the $\log d - X_{fpc}$ fitting has minimized uncertainty.

As explained above, in the new fitting exercise, we first obtain $n - X_{fpc}$ relation based on both continuous and discontinuous models (Fig. 3D). Then we obtained two sets of k from $d^n = kt$ (Ext. Data Table 2; Fig. 3E). Finally, we calculated d as a function of X_{fpc} (Fig. 3A-C) and simulated the viscosity based on both the continuous and discontinuous models (Fig. 4).

2) The authors’ disregard of analytical/micromechanical models for grain growth appears in stark contrast to their “uncritical belief” in creep laws (eqs. 3 and 4 in appendix). The manuscript lacks a critical discussion of either the model equations or the used parameters (and their uncertainties). This lack becomes critically important at the point of the “logical thread”, where the authors realize that the predictions for a potential viscosity jump actually critically depend on the (quantitative predictions regarding) dislocation creep behavior.

The uncertainty analysis about the simulation models is given in Line 415-442. The viscosity contrast and the contribution of dislocation creep is explained in more detail (Line 121-129; Line 415-432). For the model equations, since measurements of creep laws under lower mantle conditions is technically unrealistic, Eq. 3 and 4 are currently most reliable and commonly used models for simulation of bridgmanite creep laws (e.g. Xu et al. 2011; Boioli et al. 2017; Reali et al. 2019). Clearly there are uncertainties in these creep laws, but they do not affect our conclusion that bridgmanite-enriched rocks are more than one order of magnitude stronger than pyrolitic rocks.

The major uncertainty in Eq. 3 and 4 comes from the error bars of D^{lat} and δD^{gb} . The viscosity contrast is indeed the ratio of creep rates between pyrolitic rocks and bridgmanite-enriched rocks. When $d \gg 1 \mu\text{m}$, we have $D^{\text{lat}} \gg \delta D^{\text{gb}}/d$ (Yamazaki et al. 2000). Both ϵ_{diff} and ϵ_{dis} become linearly proportional to D^{lat} . Therefore, the ratio of creep rates between pyrolitic rocks and bridgmanite-enriched rocks becomes independent of D^{lat} and δD^{gb} . The uncertainties of D^{lat} and δD^{gb} (and their P, T dependences as well) will not affect the viscosity contrast between bridgmanite-enriched and pyrolitic rocks. They only affect the absolute values of the simulated η (Line 416-427).

We agree the viscosity contrast relies on the contribution of dislocation creep in bridgmanite-enriched rocks as pointed out by reviewer #2. The bridgmanite-enriched rocks are dominated by either diffusion creep or dislocation creep (upon X_{fpc}). When dislocation creep dominates, the ratio of creep

rates between the two rocks is $\left(\frac{1}{d\sigma}\right)^2 \frac{\pi AG^2 \ln(4G/\pi\sigma)}{b}$. If diffusion creep dominates in the bridgmanite-enriched rocks, the ratio is $\left(\frac{1}{d}\right)^2$. Therefore, in both cases the viscosity contrast is significantly controlled by grain size d . Since shear modulus G and Burgers vector b have negligible uncertainties compared with the magnitude of viscosity contrast, the ratio of creep rates between the two rocks is only strongly controlled by grain size d and stress σ . With d from this study and $\sigma \leq 1$ MPa, the grain-size contrast will always (i.e., within uncertainties) induce a viscosity contrast by more than one order of magnitude even if dislocation creep dominates in bridgmanite-enriched rocks (Fig. 4C). The only way to reduce the viscosity contrast would be a higher stress ($\gg 1$ MPa), however, the stress in the general area of Earth's mantle is ≤ 1 MPa as estimated from the velocities of upwelling and downwelling flows (e.g. Karato, 2008) (Line 419-432).

The role of pressure on diffusion kinetics should, for example, be addressed. What kind of activation volume would be required to observe a change from “thermal activation” to “pressure deactivation”.

Because the activation volume for diffusion (ΔV) in bridgmanite is unknown, we assume different values 0, 1, 2, and 3 cm³/mol. The simulated viscosity-depth profile is given in Ext. Data Fig. 3.

As shown in the figure, ΔV affects the slope of the viscosity – depth profile. When ΔV is 0 – 1 cm³/mol, the viscosity has small negative depth dependence. When ΔV is 1 – 3 cm³/mol, the viscosity has small positive depth dependence. $\Delta V > 3$ cm³/mol is unlikely because the viscosity will increase by more than four orders of magnitude in the lower mantle, which is against the geoid observation.

However, no matter which value is assumed (from 0 to 3 cm³/mol), ΔV does not affect the viscosity jump at ~1000 km depth. This is reasonable because D_{Si} varies continuously with depth since the pressure and temperature vary with depth continuously (Line 433-442, Ext. Data Fig. 3).

reasoning

As intriguing as grain size may be as the cause for the viscosity jump, it brings a few problems with it. Grain size is an “unbounded” internal variable (as long as dynamic recrystallization, irreversible thermodynamics, is excluded, see below); the static thermodynamic equilibrium of the mantle corresponds to single crystals of the involved phases. Thus, one has to answer the central question why the current micro-structural state is such that the viscosity jump results. In this regard, the manuscript remains shallow. I cannot identify convincing arguments why the ferropericlas concentration should vary with depth in the required form (it is this content that in the authors’ reasoning controls grain size).

The X_{fpc} variation in the lower mantle is discussed in more detail in Line 38-43.

Compare to the shallow regions with pyrolitic composition, bridgmanite-enriched rocks could be formed from a deep magma ocean in the deep part of lower mantle and preserved until the present-day, as supported by geophysical, geochemical, and geodynamic evidences (e.g., Murakami et al. Nature 2012; Mashino et al., PNAS 2020; Ballmer et al. Sci. Adv. 2015; Nat. Geo 2017), and also supported by the higher Mg/Si in the upper mantle than the Earth's building blocks (chondrites) (e.g., Allegre et al., EPSL 1995).

The finding of this study is that such bridgmanite-enriched rocks in the deep lower mantle would evolve to much larger grain sizes than the overlying pyrolitic rocks. For example after 1 Gyr, the grain size of pyrolitic rocks is $\sim 300 \mu\text{m}$, but that of bridgmanite-enriched rocks will be $\sim 30 \text{ mm}$ (Fig. 4A, upon X_{fpc}). This grain size contrast will cause a large viscosity contrast, which well explains the observed viscosity increase at 800 – 1200 km depth.

The authors neglect that the particles' content is only one part of the role of secondary phases on grain growth. Actually, interparticle spacing controls the spatial length for unhindered growth. Quantification of interparticle spacing requires at list the incorporation of particle size in addition to content. Thus, interparticle spacing should be determined and reported. The authors' conceptual focus on "particle content" leads to the critical omission of Ostwald ripening of secondary particles. The interparticle spacing of ferropericlas in the mantle, the critical length scale for grain size of bridgmanite, is not discussed whatsoever. Four billion years is a long time even for a process that one might consider "slow" –on whatever scale.

Following the suggestions, the quantified interparticle spacing values are reported, and the related discussion and figure are added (Line 105-111; Ext. Data Fig. 2). However, not all the experimental data points in this study can be described by interparticle spacing. That is because based on the definition, the interparticle spacing becomes invalid for the samples with $X_{fpc} = 0$, and becomes meaningless for samples with high X_{fpc} in which ferropericlas grains are significantly interconnected.

As shown in Ext. Data Fig. 2, the grain size of bridgmanite is controlled by interparticle spacing, while the interparticle spacing is controlled by X_{fpc} and duration. Therefore, the $d - X_{fpc} - t$ relation can be a reasonable approaching to describe our experimental data. The increase of interparticle spacing with duration (Ext. Data Fig. 2B) indicate the Ostwald ripening of X_{fpc} during grain growth experiments as suggested by the reviewer.

As the authors realize, whether grain size can account for a viscosity jump critically depends on the potential to activate deformation mechanisms alternative to diffusion creep. Dislocation creep limits the grain-size effect. Grain-growth kinetics alone do not allow one to make predictions on viscosity variations –as apparently "implied" by the study- but all reasoning regarding viscosity critically hinges on the dislocation-creep law, outside of the "research" of the authors. To some extent, the false impression is created that studying grain growth provides the rheology of the mantle. Quantitative estimates of viscosity for the mantle hinge on constraints on flow laws rather than on anything constrained by this study alone. In addition to these conceptual issues, the relevant processes may change when dislocation creep is active activation. When operating for sufficiently large strains, dislocation creep is associated with "dynamic recrystallization". Then, the grain size is not controlled by the growth law alone but grain size may assume a "steady-state" value dictated by the power exerted on the mantle volume.

The contribution of dislocation creep is explained in more detail in Line 121-129 and Line 415-432. Also see our replies above.

We agree that the viscosity contrast relies on dislocation creep. As explained above, when dislocation creep dominates in bridgmanite-enriched rocks, the viscosity contrast is indeed the ratio $\left(\frac{1}{d\sigma}\right)^2 \frac{\pi A G^2 \ln(A G / \pi \sigma)}{b}$, which is significantly controlled by grain size (Line 421-432).

The grain-size contrast by itself causes the diffusion-creep rate of pyrolitic rocks to be more than four orders of magnitude higher than in bridgmanite-enriched rocks. By adding the grain-size independent dislocation creep rate, the total creep rate of pyrolitic rocks remains 1 – 2 orders of magnitude higher (Fig. 4B; 4C). Therefore, the viscosity contrast is still mainly caused by the grain-size contrast even if dislocation creep dominates in bridgmanite-enriched rocks (Line 121-129).

In the terms of large strains, as we explained in the manuscript (Line 167-170), mantle flow does not occur significantly in the bridgmanite-enriched regions (deep lower mantle) as has been shown by geodynamic modelling (e.g. Ballmer et al., 2015; 2017). Therefore, dynamic recrystallization should be inefficient and grain size reduction by dynamic recrystallization is not expected. Large strains could be localized only at the margins of the bridgmanite-enriched regions such as near subducting slab.

writing:

Please see the digital annotations indicating where I think the wording lacks conciseness.

All the suggestions and corrections in the digital files are accordingly revised. Because they are relatively minor corrections, we did not list them here, but we keep all the tracked changes in the revised manuscript.

formal aspects:

To me, the structure of the manuscript is somewhat odd. The authors seem to spend more “time”, i.e., more text, on discarding alternative reasons for changes in viscosity than for the grain-size related ones advocated by them. Actually, the discard of previously presented alternative, to me, requires a concise presentation in the introduction, rather than in the discussion, as part of the “research-gap identification” motivating the study of grain-growth kinetics.

Following the suggestions, the related text about previous alternatives is moved to the introduction. They are improved for conciseness (Line 46-53).

Reviewer Reports on the First Revision:

Referee #1:

Review comments on a revised manuscript of "Bridgmanite grain size variation accounts for the mid-mantle viscosity jump" by Hongzhan Fei et al. submitted to Nature

This manuscript reports a study of effect on ferropericlase content on grain-growth in bridgmanite-ferropericlase aggregate based on high-pressure and high-temperature experiments. Based on the experimental results, the authors discuss variation of grain-size and viscosity in the Earth's lower mantle. The discussion is original and of broad interests.

The revised manuscript includes a large number of additional experimental data which were taken using glass starting material. Based on the new data, grain-growth law in bridgmanite-ferropericlase aggregates is determined confidently as a function of ferropericlase volume fraction. The issues pointed out for the original manuscript were adequately revised in this revised version, and quality of manuscript was improved significantly. I only have a trivial comment as described below.

Although presence of another phase than bridgmanite and ferropericlase is evident in some run products from sol-gel starting materials (see brightest phase in Fig. S49-53), this is not described at all. I suspect this phase is metallic iron. This observation and its influence on grain-growth should be described in main text or method section.

Referee #2 (Joerg Renner):

You asked me to review a revised version of the manuscript entitled "Bridgmanite grain size variation accounts for the mid-mantle viscosity jump" authored by Hongzhan Fei, Maxim D. Ballmer, Ulrich Faul, Nicolas Walte, and Tomoo Katsura submitted to nature communications. Here, I am more than happy to acknowledge the authors' effort in improving the data analysis. However, my fundamental reservations about the implications, they think their work has, remain (see the itemized comments below). In addition, the writing exhibits a tendency of progressing from "could be" to "is". To me, already the abstract is a good example of the omission of the important "ifs" that apply to the implications the authors wish to deduce from their study. Then, there is a tendency for "convolution"; an example (line 39):

"... the shallow part of the lower mantle is considered to be pyrolytic with a relatively high X_{fpc} (~20%), whereas bridgmanite-enriched rocks with low X_{fpc} (< 5 – 10%) may have formed in the early stage of Earth history by fractional crystallization of the magma ocean¹³ and be preserved in the deeper regions against mantle convection as indicated by shear-wave velocity, density, and geodynamic modelling^{5-7,14-16}."

The authors should present the "evidence" for the current (compositional) state of the Earth's interior first, i.e., shear-wave velocity and density determinations –and, to a lesser degree, geodynamical modeling, before speculating on the origin of the compositional state.

As before ... Please, find detailed comments below and attached a digitally annotated pdf-version of the manuscript. My comments in the pdf-file reflect my immediate responses when reading the manuscript and did not experience any "post"-polishing for "diplomacy" or the like; they may thus appear frank in cases but no offense meant whatsoever.

Prominently, three problems are still insufficiently addressed:

1) ferropericlase content in the lower mantle: What are reliable constraints? (seismic velocities, density) In particular, in case reliable constraints do not exist, it is necessary to have a plausible explanation why ferropericlase content should vary systematically in the lower mantle. Without

this depth-dependence of ferropericlasite content, the entire argument about (related) grain-size variations is void.

Just a weird suggestion: Maybe, buoyancy provides an answer. If there was any heterogeneity in ferropericlasite content (by chance upon formation of the mantle) early on, density contrasts may have furthered their "vertical" separation. Not my expertise at all ... but if I understand the study by Ricolleua et al GRL 2009 (doi:10.1029/2008GL036759) correctly, fpc switches from lower to higher density than bridgmanite with increasing pressure.

2) the role of dislocation creep: The dislocation creep law for bridgmanite aggregates with ferropericlasite as second phase has, to my understanding, not been experimentally determined. Thus, the authors have to rely on "guesstimates" as plausible as they may think they are. In fact, rapid grain growth "works against" their reasoning. At fixed stress, dislocation creep will eventually become the rate-controlling process when grains become too large. If the bridgmanite grains had four billion years for their growth ...

3) the role of Ostwald ripening: The authors' reasoning regarding "interparticle spacing" and its role for their kinetics laws is unclear to me. If Ostwald ripening was the faster of the two processes, coarsening of bridgmanite matrix and coarsening of the "isolated" ferropericlasite second-phase, the grain-growth kinetics should not depend on ferropericlasite content, at least for relatively dilute mixtures, say up to 10 - 20% second phase. If Ostwald ripening is relatively slow compared to matrix coarsening, the interparticle spacing constitutes the upper limit for bridgmanite grain size. In this case, one cannot make any predictions on grain size in the matrix without knowing the initial interparticle spacing and the ripening law. Qualitatively, a decrease in second-phase content with depth would actually favor an increase in grain size of the bridgmanite component with depth, i.e., what the authors would like it to be.

*****END*****

Author Rebuttals to First Revision:

A list of responses to reviewers' comments

We sincerely appreciate the anonymous reviewer and Prof. Jörg Renner for the helpful comments.

We tried our best to address all the reviewers' concerns and to improve the manuscript following the suggestions. Below is a detail list of our response to reviewers' comments. In case if we have misunderstood any comments from the reviewers, we would be happy to revise the manuscript by further suggestions.

=====

Referee #1:

Review comments on a revised manuscript of "Bridgmanite grain size variation accounts for the mid-mantle viscosity jump" by Hongzhan Fei et al. submitted to Nature

This manuscript reports a study of effect on ferropericlase content on grain-growth in bridgmanite-ferropericlase aggregate based on high-pressure and high-temperature experiments. Based on the experimental results, the authors discuss variation of grain-size and viscosity in the Earth's lower mantle. The discussion is original and of broad interests.

The revised manuscript includes a large number of additional experimental data which were taken using glass starting material. Based on the new data, grain-growth law in bridgmanite-ferropericlase aggregates is determined confidently as a function of ferropericlase volume fraction. The issues pointed out for the original manuscript were adequately revised in this revised version, and quality of manuscript was improved significantly. I only have a trivial comment as described below.

Although presence of another phase than bridgmanite and ferropericlase is evident in some run products from sol-gel starting materials (see brightest phase in Fig. S49-53), this is not described at all. **I suspect this phase is metallic iron. This observation and its influence on grain-growth should be described in main text or method section.**

It is metallic iron in Fig. S49-53. It is expected to have negligible effect on the $\log d - X_{fpc}$ relation because the volume fraction of the metallic iron is very small in comparison with X_{fpc} in S49-53. This negligible effect is also confirmed by the consistent results in the different samples without metallic iron (Line 415-417).

Referee #2 (Joerg Renner):

You asked me to review a revised version of the manuscript entitled "Bridgmanite grain size variation accounts for the mid-mantle viscosity jump" authored by Hongzhan Fei, Maxim D. Ballmer, Ulrich Faul, Nicolas Walte, and Tomoo Katsura submitted to nature communications. Here, I am more than happy to acknowledge the authors' effort in improving the data analysis. However, **my fundamental reservations about the implications**, they think their work has, remain (see the itemized comments below).

Our explanations to referee #2's concerns are shown below.

In addition, the writing exhibits a tendency of progressing from "could be" to "is". To me, already the abstract is a good example of the omission of the important "ifs" that apply to the implications the authors wish to deduce from their study. Then, there is a tendency for "convolution"; an example (line 39):

" ... the shallow part of the lower mantle is considered to be pyrolitic with a relatively high X_{fpc} (~20%), whereas bridgmanite-enriched rocks with low X_{fpc} (< 5 – 10%) may have formed in the early stage of Earth

history by fractional crystallization of the magma ocean¹³ and be preserved in the deeper regions against mantle convection as indicated by shear-wave velocity, density, and geodynamic modelling^{5-7,14-16}.”

The authors should present the “evidence” for the current (compositional) state of the Earth’s interior first, i.e., shear-wave velocity and density determinations –and, to a lesser degree, geodynamical modeling, before speculating on the origin of the compositional state.

In the revised manuscript we tried to revise the text to avoid the tendency of “could be” or “considered to be” to “is” and to avoid the “convolution”.

Evidences of the compositional state of the mantle is explained below and in Line 34-43 of the manuscript.

As before ... **Please, find detailed comments below and attached a digitally annotated pdf-version of the manuscript.** My comments in the pdf-file reflect my immediate responses when reading the manuscript and did not experience any “post”-polishing for “diplomacy” or the like; they may thus appear frank in cases but no offense meant whatsoever.

All the suggestions and corrections in the digital files are accordingly revised. Because they are relatively minor corrections about re-wording and re-organizing of text, we did not list them here, but we keep the tracked changes in the revised manuscript.

Prominently, three problems are still insufficiently addressed:

1) ferropericlase content in the lower mantle: **What are reliable constraints?** (seismic velocities, density) In particular, in case reliable constraints do not exist, **it is necessary to have a plausible explanation why ferropericlase content should vary systematically in the lower mantle.** Without this depth-dependence of ferropericlase content, the entire argument about (related) grain-size variations is void.

We agree that the variation of X_{fpc} with depth is a hypothesis given by previous studies and our implication is based on this hypothesis. However, we point out that this hypothesis is rather reliable because there are a series of mineral-physics experiments, seismic velocity, density, and geodynamic modelling evidences that support this hypothesis as explained below and in Line 34-43 of the manuscript.

Experiments of silicate melting and solidification (e.g., Fiquet et al., Science 2010; Walter et al., GCA 2004; Ito et al., PEPI 2004; Nabiei et al., GRL 2021) demonstrate that bridgmanite is the first phase to crystallize in the deep lower mantle at >35 GPa. Fractional crystallization during magma ocean solidification causes bridgmanite-enriched rocks in the deep lower mantle (>~1000 km depth) and pyrolitic (or peridotitic) rocks in the shallow regions (Xie et al., Nature Com. 2020). Geodynamic modelling (Ballmer et al., Nature Geo. 2017; Gülcher et al., EPSL 2020; 2021) suggest that the early crystallized bridgmanite-enriched rocks should be preserved until the present-day. This is demonstrated by the present-day seismic profiles: the shear-wave velocity and density of bridgmanite-enriched rocks agrees well with mantle seismic profiles, but those of pyrolitic-rocks do not agree with seismic profiles (Murakami et al., Nature 2012; Mashino et al., PNAS 2020; Ricolleua et al., 2009). Additionally, pyrolitic rocks are gravitationally stable in the shallow regions because of the lower density of ferropericlase than bridgmanite at low pressures (Ricolleua et al., 2009).

Just a weird suggestion: Maybe, buoyancy provides an answer. If there was any heterogeneity in ferropericlase content (by chance upon formation of the mantle) early on, density contrasts may have furthered their “vertical” separation. Not my expertise at all ... but it if I understand the study by Ricolleua et al GRL 2009 (doi:10.1029/2008GL036759) correctly, fpc switches from lower to higher density than bridgmanite with increasing pressure.

Actually, as argued in the final paragraph of Ricolleua et al. (2009), their density data suggests bridgmanite-enriched rocks in the lower mantle.

As explained above, the variation of X_{fpc} with depth is not formed “by-chance”, but is a well-constrained hypothesis based on a series of evidences. The bridgmanite enriched rocks are formed

in the deep lower mantle by solidification of the magma ocean. The pyrolitic rocks are gravitationally stable in the shallow regions because at low pressures ferropericlase has lower density than bridgmanite (Ricolleua et al., 2009).

(Line 34-43)

2) the role of dislocation creep: The dislocation creep law for bridgmanite aggregates with ferropericlase as second phase has, to my understanding, not been experimentally determined. Thus, the authors have to rely on “guestimates” as plausible as they may think they are.

It is true that direct deformation experiments of bridgmanite is difficult and the rheologic data has large uncertainties. However, our flow laws for bridgmanite (Eq. 2, 3) are based on well-established principles about the relations between creep rate, grain size, and stress.

In fact, rapid grain growth “works against” their reasoning. At fixed stress, dislocation creep will eventually become the rate-controlling process when grains become too large. **If the bridgmanite grains had four billion years for their growth ...**

It seems that reviewer #2 misunderstood our argument about the role of dislocation creep. In fact, we exactly showed that dislocation creep dominates in bridgmanite-enriched rocks after only 100 Myr of growth.

Our argument is that, pyrolitic rocks remain in the diffusion creep regime because of the slow growth, while bridgmanite-enriched rocks are dominated by dislocation creep due to the rapid growth. The difference of viscosity between pyrolitic rocks (diffusion-creep) and bridgmanite-enriched rocks (dislocation-creep) is more than one order of magnitude, which explains the mid-mantle viscosity jump (if both are dominated by diffusion creep, the difference would be more than four orders of magnitude).

(Line 125-136, Line 444-461; Fig. 4B, 4C)

We are afraid that the audience may be concerned that dislocation creep will eventually dominate in the pyrolitic-rocks after 4.5 billion years of growth. Therefore, we revised our Fig. 4 and the related text using a timescale of 4.5 Gyr instead of 100 Myr. As shown in the figure, even after 4.5 Gyr, diffusion creep still dominates in the pyrolitic rocks, thus, the conclusion about grain-size-contrast produced viscosity-contrast is still valid (Line 117-124; Fig. 4B, 4C).

3) the role of Ostwald ripening: The authors’ reasoning regarding “interparticle spacing” and its role for their kinetics laws is unclear to me. If Ostwald ripening was the faster of the two processes, coarsening of bridgmanite matrix and coarsening of the “isolated” ferropericlase second-phase, **the grain-growth kinetics should not depend on ferropericlase content, at least for relatively dilute mixtures**, say up to 10 – 20% second phase. If Ostwald ripening is relatively slow compared to matrix coarsening, the interparticle spacing constitutes the upper limit for bridgmanite grain size. In this case, one cannot make any predictions on grain size in the matrix without knowing the initial interparticle spacing and the ripening law. Qualitatively, a decrease in second-phase content with depth would actually favor an increase in grain size of the bridgmanite component with depth, i.e., what the authors would like it to be.

The role of Ostwald ripening is explained in Line 105-116 and Extended Data Fig. 2, 3.

If we understand correctly, reviewer #2’s argument is as follows: If the fpc coarsening (Ostwald ripening) is much slower than the brg growth, brg cannot grow continuously with time because brg growth is limited by a constant value of the fpc interparticle spacing. If fpc coarsening is faster than brg growth, a small amount of fpc will not affect the brg growth.

Here, we point out that, growth of the matrix and coarsening of the second phase are theoretically two coupled processes occurring simultaneously (e.g., Brook, 1976). Namely, fpc coarsening rate should be comparable to brg growth. In other words:

If fpc coarsening does not occur, the fpc interparticle spacing should be constant with time. As a result, the brg growth should not occur because of the limitation by the constant interparticle spacing.

If fpc coarsening occurs, the interparticle spacing increases with time because the number of grains decreases. As a result, the brg grains grow without the limiting of fpc interparticle spacing.

In our experiments, the grain size of fpc increases with time (Ext. Data Fig. 2). Also, the interparticle spacing increases with time (Ext. Data Fig. 3B). These facts suggest the occurrence of fpc coarsening.

Additionally, the grain size of brg and the interparticle spacing show a linear relationship (Ext. Data Fig. 3A), and the fpc grain size increases with time with comparable rates as brg growth (Ext. Data Fig. 2). These data show comparable rates of brg growth and fpc coarsening. The bridgmanite growth rate determined by us therefore includes the kinetics of simultaneous coarsening of fpc. This justifies the application of our data to the lower mantle.

Herwegh et al. (2011) discussed that a second-phase affects growth of the major phase by imposing drag forces on moving grain boundaries, which counteract the driving force for growth of the major phase. They pointed out that there is no minimum volume fraction of the second phase needed for the drag force to occur. These predictions have been verified for the bridgmanite-ferropericalse system by our study and for the forsterite-enstatite system by Hiraga et al. (EPSL 2010). The Fig. 3 of our study and Fig. 4 in Hiraga et al. show that 3 – 3.5 % of second phase already significantly affects the growth of the major phase.

Reviewer Reports on the Second Revision:

Referee #2:

You asked me to review a once more revised version of the manuscript entitled "Bridgmanite grain size variation accounts for the mid-mantle viscosity jump" authored by Hongzhan Fei, Maxim D. Ballmer, Ulrich Faul, Nicolas Walte, and Tomoo Katsura submitted to nature communications. As before, I am more than willing to acknowledge the authors' efforts and their willingness to engage in the arguments brought forward in the review process. Unfortunately, I still find the line of reasoning unconvincing because it rests on two things we simply do not know (and their particular study does not contribute to learning about them):

- 1) The flow laws of lower mantle rocks.
- 2) The spatial distribution of composition variations in the lower mantle.

The authors' inference of a viscosity jump hinges on a dislocation flow law derived from theoretical considerations (Reali et al., 2019) that so far has not been substantiated by experimental constraints. Dislocation creep is a complex process and even for materials that have been extensively tested in the laboratory we can hardly agree on the underlying micromechanical laws (besides the work of Nabarro there are, for example, the laws proposed by Weertman, to name just one alternative).

Even if the particular law chosen for the analysis (based on Nabarro 1967) is the true law, the uncertainty in its parameters had to be accounted for; I do not see this thoroughly done in the current manuscript. Importantly, even without this uncertainty analysis of the flow law parameters, an account for the uncertainty of the state variables characteristic for the lower mantle alone puts the authors' conclusions into question. Only for stresses up to 1 MPa they find that diffusion creep takes over with increasing periclase content. Reali et al. (2019), the study from which the dislocation flow law is taken, however, consider 1 to 10 MPa the relevant range of stress ...

The authors' explanation for a "systematic" change in composition at mid-mantle depths does not convince me. They provide explanations for the preservation of heterogeneity in composition but not for a spatial preference for the occurrence of certain compositions: "Geodynamic modelling studies show that the bridgmanite-enriched rocks can be preserved without mixing with pyroclitic rocks until the present-day." On a side note, I find the statement "This timescale is substantially shorter than that of mantle convection and mixing." rather speculative given that we have little idea about the rheology of early Earth. (It probably should read: "This timescale is substantially shorter than that of present-day mantle convection and mixing." but then one may wonder what use it is ...)

The authors performed a great experimental study whose results deserve publication; I do not understand why they want to overstress the implications. Actually, I find their "attitude" somewhat schizophrenic. On the one hand, the grain growth data are not discussed in the light of known micro-mechanical models (grain growth and/or Ostwald ripening); even though they did not determine an activation enthalpy for growth one could look into order of magnitude correspondence between diffusion studies and their results, or discuss what is the time-limiting process grain boundary migration or ripening, etc. On the other hand, they put "blind faith" into micro-mechanical laws for creep.

We reached a point, where asking my opinion once more does not seem constructive to me. It is probably simply your call: Does nature communications foster this type of "thoughts on if ..." and if so, how should the various "ifs" involved be labeled? Regarding "careful wording" about what is speculation and what not, I cannot refrain from mentioning that the title does not leave room for interpretation but simply states that the viscosity jump is due to grain size variations.

*****END*****

Author Rebuttals to Second Revision:

A list of responses to reviewers' comments

Dear Dr. VanDecar,

First of all, we sincerely appreciate your kind reconsideration of our manuscript.

The major claim of reviewer #2 that questioned our work was that the flow law of bridgmanite and the compositional variation in the lower mantle are unknown. However, many previous studies have investigated these two issues and led to important conclusions as explained below. To clarify these issues for potential readers, the manuscript is revised accordingly (**Line 34-45; Line 431-474; Extended Data Fig. 4; Line 561-572**).

Referee #2:

You asked me to review a once more revised version of the manuscript entitled "Bridgmanite grain size variation accounts for the mid-mantle viscosity jump" authored by Hongzhan Fei, Maxim D. Ballmer, Ulrich Faul, Nicolas Walte, and Tomoo Katsura submitted to nature communications. As before, I am more than willing to acknowledge the authors' efforts and their willingness to engage in the arguments brought forward in the review process. Unfortunately, I still find the line of reasoning unconvincing because it rests on two things we simply do not know (and their particular study does not contribute to learning about them):

- 1) The flow laws of lower mantle rocks.
- 2) The spatial distribution of composition variations in the lower mantle.

About the flow law:

We used theoretical equations to calculate the creep rates of bridgmanite, which reviewer #2 considers unacceptable. However, the validity of these equations has already been demonstrated by other studies and they have been commonly used in the community of mantle rheology, especially for the deep mantle.

The theoretical equations we used are:

$$\dot{\epsilon}_{\text{diffusion}} = A \frac{\sigma V_m}{RT a^2} \left(D^{\text{lat}} + \frac{\delta D^{\text{gb}}}{a} \right) \quad (\text{Eq. 2 in Method section})$$

$$\dot{\epsilon}_{\text{dislocation}} = \frac{D^{\text{lat}} b \sigma^3 V_m}{\pi RT G^2} / \ln \left(\frac{4G}{\pi \sigma} \right) \quad (\text{Eq. 3 in Method section})$$

These are well-established equations for diffusion creep and dislocation creep, respectively, in minerals and ceramic materials.

Since the experimental work on bridgmanite deformation was limited, Prof. Patrick Cordier's group spent tremendous efforts to investigate the flow law of bridgmanite by numerical modelling (e.g., Reali et al., Sci. Rept. 2019; PEPI 2019; Ritterbex et al., EPSL 2020; Boioli et al., Sci. Adv. 2017; Kraych et al., EPSL 2016; PRB 2016...). They concluded that dislocation creep of bridgmanite is controlled by pure climb mechanism, hence, Eq. (3) is suitable to describe the dislocation creep rate in bridgmanite. On the other hand, Eq. (2) well describes the diffusion creep of bridgmanite.

Experimentally, Tsujino et al. (Sci. Adv. 2022) reported the stress-creep rate relationship at 1473 – 1673 K in the dislocation creep regime. Within uncertainty, their results are consistent with those calculated from Eq. (3). Direct deformation experiments in the diffusion creep regime of bridgmanite are impractical at present, but the validity of Eq. (2) has been experimentally tested on other rock-forming minerals such as olivine and pyroxene as shown below (Yabe et al., 2020; Faul & Jackson, 2007; Ghosh et al., 2021; Tasaka et al., 2013). Therefore, theoretical calculations from Eq. (2, 3) should give a reasonable estimation of creep rates in bridgmanite.

Consistency of dislocation creep rate in bridgmanite calculated from Eq. (3) and from deformation experiments by Tsujino et al. (2022).

[REDACTED]

Consistency of diffusion creep in olivine calculated from Eq. (2) and obtained from deformation experiments. Left figure is from Yabe et al. (2020). Symbols: from calculation. Solid lines: from deformation experiments. Right figure is from Faul & Jackson (2007). Symbols: from deformation experiments. Solid lines: from calculation.

Consistency of diffusion creep in pyroxene from calculations and from deformation experiments adjusted to a stress of 30 MPa and grain size of 1 μm (equivalent to the experimental conditions in Ghosh et al. 2021 and Tasaka et al. 2013). Note that Ghosh et al. (2021) concluded that the diffusion creep rates in diopside between calculations and experiments are inconsistent, however, they only calculated the Nabarro-Herring creep regime, while the Coble creep regime is not negligible in their small-grain-size samples ($\sim 1 \mu\text{m}$). By considering the Coble creep, the simulation and deformation experiments become consistent.

We also emphasize that using the above theoretical models is a widely accepted approach to investigate mantle rheology, especially for the deep mantle (e.g., the above publications from Prof. Cordier's group; Xu et al., JGR 2011; Shimojuku et al., EPSL 2009; Tsujino et al., Sci. Adv. 2022, and many others). Of course, the models contain uncertainties. However, as explained in the Method section, these uncertainties do not affect our conclusions. Even though the experimentally determined Si diffusivity (D) has a relatively large uncertainty, D is cancelled in calculating the viscosity contrast between bridgmanite-enriched rocks and pyrolitic rocks ($\delta D^{\text{gb}}/d \ll D^{\text{lat}}$). The other parameters, such as shear modulus G , molar volume V_m , and burgers vector b are all reasonably well constrained. The stress (σ) in the mantle has been estimated to be $\ll 1$ MPa in previous studies. The grain size is constrained in this study.

The related explanation about the flow law is given in **Line 431-474** and in **Extended Data Fig. 4** of the manuscript.

About the composition variation in the lower mantle:

The essential compositional parameter for the current study is the lower X_{fpc} (bridgmanite-enriched) in the deep lower mantle than the shallow regions (Fig. 4D-E), which the reviewer #2 considered it is unacceptable. However, there is a broad range of evidence from various disciplines that supports bridgmanite-enrichment in the deep lower mantle.

A bridgmanite-enriched deep lower mantle is inferred from seismic evidence, i.e., the seismic profiles at the top part of the lower mantle match with pyrolitic composition, but the deeper region does not (Kurnosov et al., Nature 2017). Instead, the deep region matches with the seismic velocity and density of bridgmanite-enriched rocks (Murakami et al., Nature 2012; Mashino et al., PNAS 2020; Ricolleau et al., GRL 2009). Regarding density, pyrolitic rocks have a lower density at mid-mantle conditions than bridgmanite-enriched rocks (Ricolleau et al., 2009), which also suggests a pyrolitic shallow lower-mantle and a bridgmanite-enriched deep lower mantle.

Melting experiments demonstrate that the bridgmanite-enriched rocks should have formed during primordial magma ocean solidification because bridgmanite is the first phase to crystallize in the deep lower mantle (Fiquet et al., Science 2010; Walter et al., GCA 2004; Ito et al., PEPI 2004; Nabiei et al., GRL 2021). Fractional crystallization during magma ocean solidification produced bridgmanite-enriched rocks in the deep lower mantle ($> \sim 1000$ km depth) and pyrolitic (or peridotitic) rocks in the shallow regions (Xie et al., Nature Com. 2020).

For such an initial compositional structure of the solid mantle after magma-ocean crystallization, geodynamic modelling show that bridgmanite-enriched rocks can be preserved in mantle convection until the present day, primarily at $\sim 1000 - 2500$ km depth, as long as they are more viscous than the surrounding pyrolitic rocks (Ballmer et al., Nature Geo. 2017; Gülcher et al., EPSL 2020; Solid Earth, 2021).

Finally, the bridgmanite-enrichment in the lower mantle is also consistent with ^{142}Nd , ^3He , and ^{182}W anomalies in igneous rocks (Peters et al., Nature 2018; Mundl et al., Science 2017), broad seismic reflectors in the mid mantle (Waszek et al., Nature Comm. 2018), apparent depletion of the bulk silicate Earth compared to chondrites (Allegre et al., EPSL 1995), and so on.

All of these studies point toward a bridgmanite-enriched deep lower mantle and a pyrolitic shallow lower mantle, respectively. Therefore, the compositional model we used in the current study is reasonable.

The related explanation is given in **Line 34-45** of the manuscript.

The authors' inference of a viscosity jump hinges on a dislocation flow law derived from theoretical considerations (Reali et al., 2019) that so far has not been substantiated by experimental constraints. Dislocation creep is a complex process and even for materials that have been extensively tested in the laboratory we can hardly agree on the underlying micromechanical laws (besides the work of Nabarro there are, for example, the laws proposed by Weertman, to name just one alternative).

As explained above, the validity of Eq. (2) for dislocation creep in bridgmanite has been demonstrated both experimentally and theoretically (**Line 433-455, Extended Data Fig. 4**), and the calculated viscosity contrast between bridgmanite-enriched rocks and pyrolitic rocks already explains the viscosity jump in the lower mantle. We agree that dislocation creep is a complex process with underlying micromechanics that is not fully understood. But that is not the goal of this study and does not question our main conclusion according to the current knowledge.

About the choice of models (Nabarro vs. Weertman), this is not arbitrarily chosen as claimed by reviewer #2. The difference between the two models is the role of dislocation glide. The Nabarro equation describes a pure-climb controlled dislocation creep, while the Weertman equation describes the dislocation creep with significant contribution by dislocation. As explained above, Prof. Patrick Cordier's group showed that dislocation creep of bridgmanite under the lower mantle condition is dominated by climb mechanism (e.g., Boioli et al., 2017; Reali et al., 2019). Therefore, as far as there is no reason to discard Prof. Cordier's work, we should choose the Nabarro equation. (**Line 434**)

Even if the particular law chosen for the analysis (based on Nabarro 1967) is the true law, the uncertainty in its parameters had to be accounted for; I do not see this thoroughly done in the current manuscript.

A detailed uncertainty analysis is given in the method section both in the first and second revisions (**Line 446-485**), but somehow has been missed by reviewer #2.

As we explained in the Method section, although the experimentally determined Si diffusivity (D) in Eqs (2, 3) has a relatively large uncertainty, it does not affect our conclusions because D cancels out when calculating the viscosity contrast between bridgmanite-enriched rocks and pyrolitic rocks. The other parameters (shear modulus G , molar volume V_m , burgers vector b) are all well constrained, while the stress (σ) of <1 MPa is estimated in previous studies, and grain size is determined in this study (**Line 456-474**).

We would be happy if the reviewer could point out the uncertainty of which parameter we did not discuss sufficiently so that we can further improve it.

Importantly, even without this uncertainty analysis of the flow law parameters, an account for the uncertainty of the state variables characteristic for the lower mantle alone puts the authors' conclusions into question. Only for stresses up to 1 MPa they find that diffusion creep takes over with increasing periclase content. Reali et al. (2019), the study from which the dislocation flow law is taken, however, consider 1 to 10 MPa the relevant range of stress ...

We are afraid that reviewer #2 may have misunderstood the meaning of the stress value of 1 – 10 MPa used in Reali et al. (2019) and the 1 MPa in our manuscript.

Reali et al. (2019) calculated the creep rate at three stress conditions, i.e., 1, 5, and 10 MPa, which is the boundary condition for the mechanism transition from diffusion creep dominated to dislocation creep dominated regime. They did not estimate the stress in the mantle based on their results.

Our argument is that, at a stress of 1 MPa, the grain size contrast between bridgmanite-enriched rocks and pyrolitic rocks is sufficient to produce the viscosity jump in the lower mantle (Fig. 4C); with smaller stress, the viscosity contrast will be more significant.

About the stress in the mantle, $\sigma < 1$ MPa is a reasonable estimation based on the plate motion velocity (e.g., Karato, 2008; Kohlstedt and Hansen, 2015). Additionally, Tsujino et al. (Sci. Adv. 2022) estimated a stress of about 0.02 – 0.3 MPa in the lower mantle based on their experimental data (Line 135-137).

The authors' explanation for a "systematic" change in composition at mid-mantle depths does not convince me. They provide explanations for the preservation of heterogeneity in composition but not for a spatial preference for the occurrence of certain compositions: "Geodynamic modelling studies show that the bridgmanite-enriched rocks can be preserved without mixing with pyrolitic rocks until the present-day." On a side note, I find the statement "This timescale is substantially shorter than that of mantle convection and mixing." rather speculative given that we have little idea about the rheology of early Earth. (It probably should read: "This timescale is substantially shorter than that of present-day mantle convection and mixing." but then one may wonder what use it is ...)

As explained above, the change in composition (fpc content) is supported by previous studies. It is reported in previously about bridgmanite-enrichment in the deep lower mantle at 1000 – 2500 km depth (Line 34-45). It seems the sentence "This timescale is substantially shorter than that of mantle convection and mixing" causes some misunderstanding. To avoid the misunderstanding, we deleted this sentence. We mean that, the bridgmanite-enriched rocks are formed very rapidly at the very early stage of Earth history. They are preserved until present day as long as they have high viscosity. The grain-growth rate studied here justifies such a viscosity contrast and the preservation of the bridgmanite-enriched rocks.

The authors performed a great experimental study whose results deserve publication; I do not understand why they want to overstress the implications. Actually, I find their "attitude" somewhat schizophrenic. On the one hand, the grain growth data are not discussed in the light of known micro-mechanical models (grain growth and/or Ostwald ripening); even though they did not determine an activation enthalpy for growth one could look into order of magnitude correspondence between diffusion studies and their results, or discuss what is the time-limiting process grain boundary migration or ripening, etc. On the other hand, they put "blind faith" into micro-mechanical laws for creep.

It seems that reviewer #2 considers that an interpretation of rheology from grain size is unacceptable and "schizophrenic" even though it rests on well-established principles about relationship between grain-size and viscosity. However, it is well known that the diffusion creep rate of mineral aggregates is proportional to $d^{-2} - d^{-3}$ (Eq. 2). Namely, the grain size is a key parameter that directly controls the rheology of polycrystalline aggregates. Such a grain-size-dependent rheology has been demonstrated by many experimental studies (e.g., Fukuda et al., JGR 2017; Mei & Kohlstedt JGR 2000; Faul and Jackson, JGR 2007). Therefore, the predication of diffusion creep from grain size is reasonable.

We also point out that there are many studies that interpret mantle rheology from grain size (e.g., Mohiuddin et al., Nat. Geo 2019; Yamazaki et al., Science 1996; EPSL 2005; Hiraga et al., EPSL 2010; Nishihara et al., PEPI 2006; Karato et al., JGR 1986; Dannberg et al., G3, 2017; Mao and Zhong, GRL

2021). Namely, the interpretation of rheology from grain size is a common approach in the community of mantle rheology.

In this study we provide a comprehensive experimental data and derive a new view of the dynamics of the lower mantle. The micro-mechanical models such as grain growth and Ostwald ripening are discussed briefly in the manuscript as necessary for understanding the data and for allowing to extrapolate the data to natural conditions (**Line 107-117; Extended Data Fig. 3A, 3B**). But our main target for this study is to better understand the lower mantle rheology in the view of grain size.

We reached a point, where asking my opinion once more does not seem constructive to me. It is probably simply your call: Does nature communications foster this type of “thoughts on if ...” and if so, how should the various “ifs” involved be labeled? Regarding “careful wording” about what is speculation and what not, I cannot refrain from mentioning that the title does not leave room for interpretation but simply states that the viscosity jump is due to grain size variations.

As we repeatedly explained, both the bridgmanite flow law and the lower mantle composition are reasonably known as concluded from previous studies based on a series of evidence, and our argument is obtained from our experimental results combined with these previous conclusions. We also improved our explanation in terms of how our results (i.e., grain growth rates as a function of f_{pc}) explain the preservation of bridgmanite-enriched rocks and the mid-mantle viscosity jump.

It seems that reviewer #2 is reluctant to accept interpretations based on previous studies even though they are supported by a series of evidence. But we emphasize that, unlike the Earth’s surface and shallow crust that can be seen directly, almost all knowledge on deep mantle composition, structure, dynamics, and evolution is interpreted from experimental or computational results. Researchers in this community are either trying to collect more evidence to support or deny the existing idea, or to propose new idea based on new results.

In terms of the title, the grain-size-induced viscosity variation is our conclusion interpreted from our results. We are not trying to exclude any other possibilities, but based on our study and the current knowledge, the grain size variation evolves naturally from the expected X_{fpc} variation in the lower mantle and is sufficient to explain the viscosity jump in the lower mantle.

In summary, we argue that our experimental data are robust, our interpretations are novel and reasonable, and of high importance for understanding the Earth’s mantle in general.

Reviewer Reports on the Third Revision:

Adjudicating referee #3:

I have read the paper and the critical review. I appreciate the quality of the critical evaluation as well as the efforts of the authors to provide a reasoned response.

This manuscript presents a very careful and detailed study that aims to clarify the size of grains in the lower mantle. Grain size is a microstructural parameter that is much discussed, largely because of its strong implication in rheological behavior depending on mechanisms. In this study, the parameter considered is the fraction of ferropericlasite in a two-phase bridgmanite/ferropericlasite aggregate. It is based on an experimental approach involving multianvil experiments at high-pressure, high-temperature. The two reviewers who examined this work emphasized the high degree of expertise that underlies this type of experiments and the quality of the data produced. I agree with these conclusions.

The main limit of the present study is that experiments were conducted at 27 GPa "only", i.e. in conditions corresponding of the uppermost lower mantle. This is obviously due to experimental constraints (and the authors are among the most advanced in this field) and it is only a limitation when the authors seek to apply their results to the whole mantle.

The previous rounds of review were conducted with rigor and led the authors to strengthen their arguments. It appears that at the end of this in-depth expertise, there are no objections to the work itself, to the quality of the results and to their interpretation (although the interpretation of grain growth data remains a difficult exercise). On the other hand, the discussion stumbles over the implications that involve a field, rheology, which is currently much debated, and which is not the authors' main expertise. I must say that I do not necessarily share all the views of the authors on this point. But it must be acknowledged that they base their discussion on arguments that are based on data from the literature.

So in conclusion, and for the science, I recommend the publication of this paper.

However, I have three points that authors can take into account:

- I come back to the pressure conditions. The data presented in this paper are "punctual" with respect to the P,T conditions covered by the implications described from line 139 (i.e. the whole lower mantle!). This is not to hold this against the authors, but on the other hand, I think that this extremely important extrapolation of data should be better emphasized in the main text in order to remove any ambiguity in the reader's appreciation of the message delivered.

- As pointed out in the review of the paper, the major uncertainties come from diffusion coefficients. Just mentioning (l 436) that diffusion coefficients come from previous studies (with references) does not allow sufficient traceability of the work. I think it would be a good practice to present a table that explicitly gives the value of the diffusion coefficients (and the different parameters) used.

- Obviously, the expertise of this paper took time so that papers directly related to this work have appeared in the meantime like the one of Cordier et al. in Nature (Jan 2023). I think that in order not to give the impression of a hiatus, the authors should have the possibility to include this work in their discussion.

*****END*****

Author Rebuttals to Third Revision:

Response to Reviewer's comments

Adjudicating referee #3:

I have read the paper and the critical review. I appreciate the quality of the critical evaluation as well as the efforts of the authors to provide a reasoned response.

This manuscript presents a very careful and detailed study that aims to clarify the size of grains in the lower mantle. Grain size is a microstructural parameter that is much discussed, largely because of its strong implication in rheological behavior depending on mechanisms. In this study, the parameter considered is the fraction of ferropericlase in a two-phase bridgmanite/ferropericlase aggregate. It is based on an experimental approach involving multianvil experiments at high-pressure, high-temperature. The two reviewers who examined this work emphasized the high degree of expertise that underlies this type of experiments and the quality of the data produced. I agree with these conclusions.

The main limit of the present study is that experiments were conducted at 27 GPa "only", i.e. in conditions corresponding of the uppermost lower mantle. This is obviously due to experimental constraints (and the authors are among the most advanced in this field) and it is only a limitation when the authors seek to apply their results to the whole mantle.

The previous rounds of review were conducted with rigor and led the authors to strengthen their arguments. It appears that at the end of this in-depth expertise, there are no objections to the work itself, to the quality of the results and to their interpretation (although the interpretation of grain growth data remains a difficult exercise). On the other hand, the discussion stumbles over the implications that involve a field, rheology, which is currently much debated, and which is not the authors' main expertise. I must say that I do not necessarily share all the views of the authors on this point. But it must be acknowledged that they base their discussion on arguments that are based on data from the literature.

So in conclusion, and for the science, I recommend the publication of this paper.

We sincerely thank reviewer #3 for the positive evaluations to our manuscript.

However, I have three points that authors can take into account:

- I come back to the pressure conditions. The data presented in this paper are "punctual" with respect to the P,T conditions covered by the implications described from line139 (i.e. the whole lower mantle!). This is not to hold this against the authors, but on the other hand, I think that this extremely important extrapolation of data should be better emphasized in the main text in order to remove any ambiguity in the reader's appreciation of the message delivered.

The limited pressure condition is emphasized and the related discussion about the pressure effect is added (Line 160-164).

By considering a negative pressure dependence (Zhang and Karato, JGR 2021), the grain size of pyrolytic rocks will decrease, resulting in a decrease of viscosity with depth. In contrast, although the grain size of bridgmanite-enriched rocks will also decrease, the dominance of dislocation creep in bridgmanite-enriched

rocks causes its viscosity independent from grain size. Thus, the viscosity contrast between pyrolitic and bridgmanite-enriched rocks will be even larger.

- As pointed out in the review of the paper, the major uncertainties come from diffusion coefficients. Just mentioning (l 436) that diffusion coefficients come from previous studies (with references) does not allow sufficient traceability of the work. I think it would be a good practice to present a table that explicitly gives the value of the diffusion coefficients (and the different parameters) used.

The diffusion coefficients at 27 GPa, 2200 K, and the parameters for the pressure and temperature corrections are given in Extended Data Table 4 and clarified in Line 417-420.

As explained in Line 432-441, the uncertainty of D^{lat} and δD^{gb} does not affect the ratio of creep rates between bridgmanite-enriched rocks and pyrolitic rocks.

- Obviously, the expertise of this paper took time so that papers directly related to this work have appeared in the meantime like the one of Cordier et al. in Nature (Jan 2023). I think that in order not to give the impression of a hiatus, the authors should have the possibility to include this work in their discussion.

The recent papers (Cordier et al., Nature 2023; Xu et al., JGR 2022) are discussed in Line 49-52.

Cordier et al. (2023) suggests slower deformation rate of periclase than bridgmanite, while Xu et al. (2022) shows identical creep rate between bridgmanite and post-spinel. Therefore, both of them point to a bridgmanite-controlled lower mantle rheology, a conclusion that strengthens our argument.